# FBXO24 modulates mRNA alternative splicing and MIWI degradation and is required for normal sperm formation and male fertility

Zhiming Li[1]*, Xingping Liu[1], Yan Zhang[1], Yuanyuan Li[1], Liquan Zhou[1]*, Shuiqiao Yuan[1,2]*

[1]Institute of Reproductive Health, Tongji Medical College, Huazhong University of Science and Technology, Wuhan, China; [2]Laboratory of Animal Center, Huazhong University of Science and Technology, Wuhan, China

*For correspondence:
lzmleo@hust.edu.cn (ZL);
zhouliquan@hust.edu.cn (LZ);
shuiqiaoyuan@hust.edu.cn (SY)

Competing interest: The authors declare that no competing interests exist.

**Abstract** Spermiogenesis is a critical, post-meiotic phase of male gametogenesis, in which the proper gene expression is essential for sperm maturation. However, the underFlying molecular mechanism that controls mRNA expression in the round spermatids remains elusive. Here, we identify that FBXO24, an orphan F-box protein, is highly expressed in the testis of humans and mice and interacts with the splicing factors (SRSF2, SRSF3, and SRSF9) to modulate the gene alternative splicing in the round spermatids. Genetic mutation of FBXO24 in mice causes many abnormal splicing events in round spermatids, thus affecting a large number of critical genes related to sperm formation that were dysregulated. Further molecular and phenotypical analyses revealed that FBXO24 deficiency results in aberrant histone retention, incomplete axonemes, oversized chromatoid body, and abnormal mitochondrial coiling along sperm flagella, ultimately leading to male sterility. In addition, we discovered that FBXO24 interacts with MIWI and SCF subunits and mediates the degradation of MIWI via K48-linked polyubiquitination. Furthermore, we show that FBXO24 depletion could lead to aberrant piRNA production in testes, which suggests FBXO24 is required for normal piRNA counts. Collectively, these data demonstrate that FBXO24 is essential for sperm formation by regulating mRNA alternative splicing and MIWI degradation during spermiogenesis.

## eLife assessment

This **important** study provides insights into the role of FBXO24 in controlling spermiogenesis and male fertility in mice. The mouse models used and the data are **convincing**. This paper will interest biomedical researchers working on reproductive biology and fertility control.

## Introduction

Spermatogenesis is a dynamic but well-organized process in which germ cells develop in the seminiferous tubules of the testis. Spermatogenesis comprises three phases: mitosis of the spermatogonia, meiosis of spermatocytes, and differentiation of round spermatids into mature sperm. The third phase is also known as spermiogenesis. During spermiogenesis, round spermatids undergo many significant changes, including the loss of cytoplasm, migration of cytoplasmic organelles, formation of the acrosome and flagellum, condensation of nuclei, and reorganization of mitochondria around the sperm midpiece (*Jha et al., 2017*). In mice, spermiogenesis is subdivided into 16 steps based on changes in acrosome structure and nuclear compaction (*Ahmed and de Rooij, 2009*; *Meistrich and Hess, 2013*).

Coordinated regulation of gene expressions in the round spermatids is essential for sperm formation. However, the underlying molecular mechanism that controls mRNA expression during spermiogenesis remains elusive.

FBXO24 (F-Box Protein 24) is a member of the F-box protein family characterized by an F-box domain. The F-box proteins constitute one of the subunits of the ubiquitin protein ligase complex called SCF (SKP1-cullin-F-box). The interaction between FBXO24 and the SCF subunits is unknown. The F-box domain recruits the F-box protein to the SCF complex, and the carboxy-terminal domain is a putative protein–protein interaction domain. According to the different substrate recognition domains, F-box protein was generally divided into three subclasses: F-box/WD repeat-containing protein (FBXW), leucine-rich repeat protein (FBXL), and F-box only protein (FBXO) with/without other unknown domain (*Kipreos and Pagano, 2000*). Compared with the studies of FBXW and FBXL, the roles of most members of the FBXO subfamily have yet to be defined. FBXO24 belongs to the FBXO class. Previous studies demonstrated some FBXO proteins could regulate spermatogenesis. For example, FBXO47 has been reported to be essential for meiosis I progression in the spermatocytes of mouse testis (*Hua et al., 2019*; *Tanno et al., 2022*), and FBXO43 was associated with male infertility and teratozoospermia (*Ma et al., 2019*). In addition, FBXO24 has been reported to recognize deacetylated nucleoside diphosphate kinase A (NDPK-A) to enhance its degradation in HeLa cells (*Chen et al., 2015*). In H1299 cells, FBXO24 could promote cell proliferation by mediating ubiquitination and degradation of protein arginine methyltransferase 6 (PRMT6) (*Chen et al., 2020*). However, the molecular and biological functions of FBXO24 remain unclear, especially its role in male fertility and sperm formation.

In this study, we find that the transcripts encode FBXO24 are abundantly expressed in the testis of humans and mice. The loss function of FBXO24 in mice resulted in male sterility, with sperm possessing a collapsed mitochondrial sheath and uncompacted chromatin. We revealed that FBXO24 interacts with splicing factors (e.g., SRSF2, SRSF3, and SRSF9) to mediate gene expressions during spermiogenesis. Unexpectedly, we discovered that FBXO24 mediates MIWI ubiquitination and regulates the chromatoid body (CB) architecture and piRNA production. Together, our study demonstrated that FBXO24 plays a critical role in controlling gene expression in haploid spermatids and sperm formation during spermiogenesis.

## Results

### FBXO24 expressed in haploid spermatids during spermiogenesis

Interestingly, qPCR (Realtime quantitative PCR) analysis of multiple mouse organs showed that FBXO24 mRNA was predominantly expressed in mouse testes (*Figure 1A*). Multi-alignment and phylogenetic analysis revealed that *Fbxo24* encodes a highly conserved protein expressed in mammals, including humans and mice (*Figure 1—figure supplement 1A*). FBXO24 has two domains, F-box and regulator of chromatin condensation 1 (RCC1), which were also highly conserved in humans and mice (*Figure 1—figure supplement 1B*). Consistent with this result, transcription expression of analysis of FBXO24 in various human organs from the GE-mini database revealed FBXO24 also a testis-enriched expression in human testes (*Figure 1—figure supplement 1C*). Further qPCR analysis of the testis at different developmental stages and purified testicular cells showed that FBXO24 mRNA was highly expressed in the round spermatids and elongating spermatids (*Figure 1B, C*). We used the CRISPR/Cas9-mediated genome-editing system to generate a mouse model containing the *Fbxo24* gene with a C-terminal HA epitope tag (*Liu et al., 2019*). To examine the protein level of FBXO24 in testes, we then characterized the tissue expression pattern of FBXO24 in various organs from adult *Fbxo24*-HA-tagged transgenic mice (*Figure 1—figure supplement 1D*). Consistent with the mRNA expression showing a testis-enriched expression of FBXO24, FBXO24[HA-Tag] protein was exclusively expressed in the testes (*Figure 1D*), suggesting FBXO24 plays a role in sperm formation. To explore the subcellular localization of FBXO24 in male germ cells, we performed immunofluorescence analysis by co-staining of FBXO24[HA-Tag] protein with γ-H2AX (a marker of meiotic DNA damage response) or PNA (an acrosome marker) in adult *Fbxo24*-HA-Tagged transgenic mouse testes. The results showed that FBXO24[HA-Tag] protein was exclusively localized in the nuclei and cytoplasm of spermatids at stages VI–IX (*Figure 1E, F*), demonstrating FBXO24 expressed in the round and elongating spermatids at

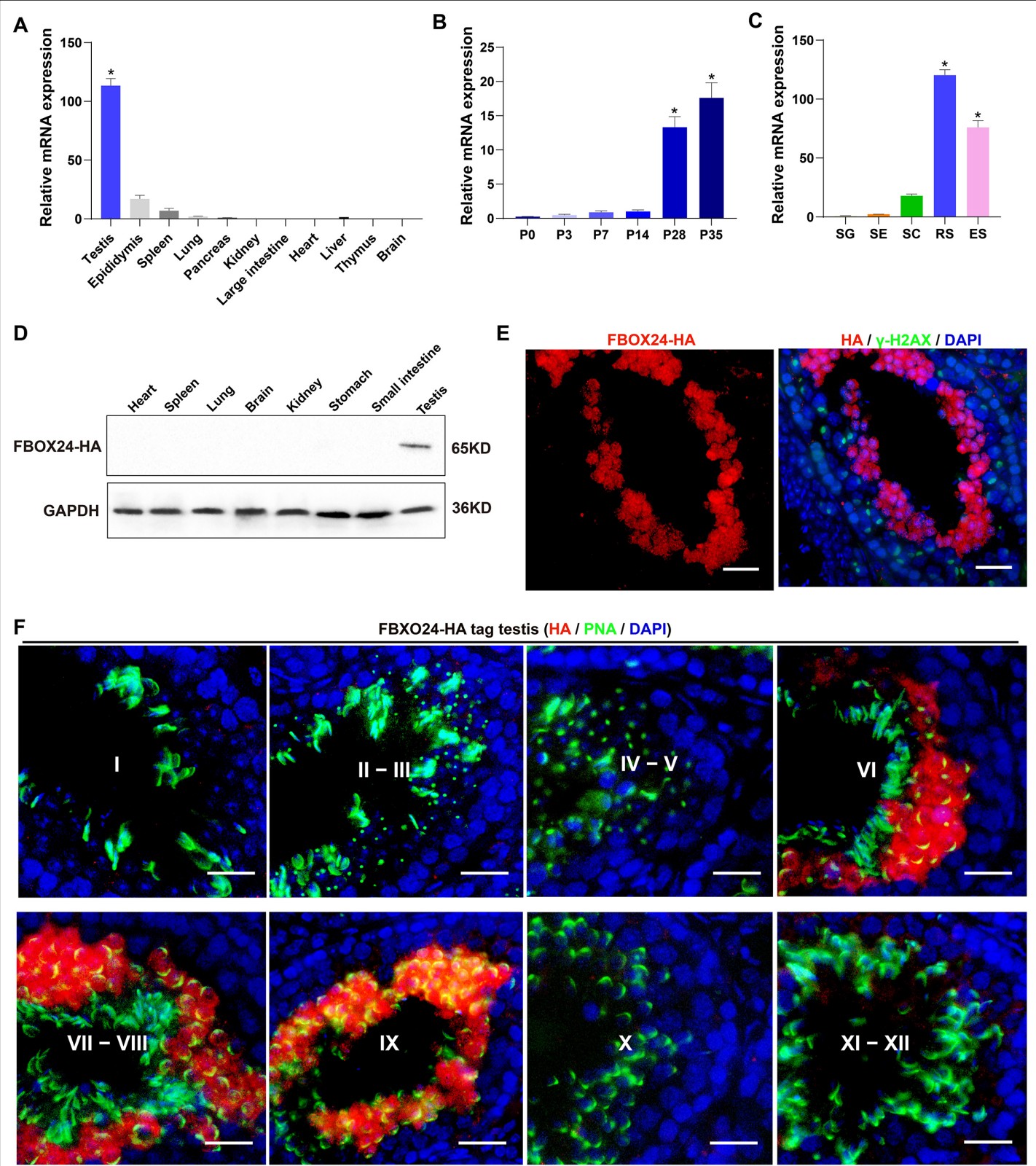

**Figure 1.** Expression profiles of FBXO24 during testicular development and spermatogenesis in mice. (**A**) qPCR analysis of FBXO24 mRNA levels in multiple organs in mice. *n* = 3/group. Data are mean ± standard deviation (SD). *p < 0.05. (**B**) qPCR analysis of FBXO24 mRNA levels in developing testes at postnatal day 0 (P0), P3, P7, P14, P28, and P35. *n* = 3/group. Data are mean ± SD. *p < 0.05. (**C**) qPCR analysis of FBXO24 mRNA levels in isolated spermatogenic cell populations, including spermatogonia (SG), Sertoli cells (SE), spermatocytes (SC), round spermatids (RS), and elongating

*Figure 1 continued on next page*

*Figure 1 continued*

spermatids (ES). *n* = 3/group. Data are mean ± SD. *p < 0.05. (**D**) Western blot analysis of FBXO24-HA expression in the tissues of adult transgenic mice. (**E–F**) Co-immunostaining of FBXO24-HA red) and (**E**) γH2AX (green) or (**F**) PNA (green) in adult *Fbxo24*-HA testes. Scale bars = 25 μm. DNA (blue) is stained by DAPI (4,'6-Diamidin-2-phenylindole dihydrochloride). Spermatogenic stages are noted.

The online version of this article includes the following source data and figure supplement(s) for figure 1:

**Source data 1.** Raw western blot for *Figure 1D*.

**Figure supplement 1.** Expression profiles of FBXO24 in mice and human.

steps 6–9. These results indicate that FBXO24 is an evolutionarily conserved testis-enriched protein specifically expressed in haploid spermatids during spermiogenesis.

## FBXO24 deficiency results in defective spermiogenesis and male infertility in mice

To reveal the role of FBXO24 in male germ cell development and sperm formation, we generated *Fbxo24* knockout (KO) mice using CRISPR/Cas9 gene editing technology. Two small guide RNAs (sgRNAs) were designed to target exon 3, which encodes the conserved F-box domain. To validate nucleotide deletion at the targeted loci, the corresponding genomic regions of *Fbxo24* were evaluated by PCR. qPCR analysis verified the mRNA expression of *Fbxo24* was almost undetectable in *Fbxo24* KO mouse testes (*Figure 2A*; *Figure 2—figure supplement 1A,B*). To test the fecundity of *Fbxo24* KO mice, we crossed *Fbxo24* KO mice with fertility-proved wild-type (WT) mice for at least 6 months. The fertility test results showed that *Fbxo24* KO males were completely infertile, but the *Fbxo24* KO females were fertile (*Figure 2B*). Interestingly, we observed the testis size and the ratio of testis weight to body weight were comparable between *Fbxo24* KO males and controls (*Figure 2C*; *Figure 2—figure supplement 1C*). Consistent with these observations, histological analysis of adult testes and epididymis showed no obvious abnormalities in *Fbxo24* KO mice compared with WT (*Figure 2—figure supplement 1D*). However, in *Fbxo24* KO mice, we found fewer elongating spermatids (at stages IX–X) and condensing spermatids (at stage XII) and a reduced number of condensed spermatids (at stages II–VIII) (*Figure 2D,E*), concomitant with increased apoptotic signals detected in late spermatids (*Figure 2—figure supplement 1E,F*), which suggesting FBXO24 deficiency could affect spermatid development during spermiogenesis. To further demonstrate the critical function of FBXO24 during spermiogenesis, we attempted to use the *Fbxo24*-HA-tagged transgenic mice to rescue sperm formation and fertility in *Fbxo24* KO males. As expected, we found that expression of the transgenic FBXO24 could rescue the defects in spermiogenesis and infertility observed in *Fbxo24* KO male mice (*Figure 2F*; *Figure 2—figure supplement 1G*), which confirmed the specific targeting of *Fbxo24* in the mutants and the true role of FBXO24 in spermiogenesis. Together, these data indicate that depletion of FBXO24 causes defective spermiogenesis, leading to male infertility.

## Ablation of FBXO24 in mice causes disorganized mitochondrial sheath of spermatozoa

To elucidate the causes of sterility in *Fbxo24* KO males, we examined sperm count, sperm motility, and sperm morphology in *Fbxo24* KO mice. The results showed that the sperm count and motility were significantly reduced in *Fbxo24* KO mice compared with WT (*Figure 3A,B*). Inspiringly, we found that most of the spermatozoa in *Fbxo24* KO mice displayed a disorganized mitochondrial sheath and had abnormal bending of the flagella (*Figure 3C*). In WT spermatozoa, the midpiece connected smoothly to the principal piece; however, in *Fbxo24* KO spermatozoa, a gap in the midpiece can be observed (*Figure 3C,D*), leading to many sperm with an abrupt bending of the tail in the region of the neck and midpiece (*Figure 3—figure supplement 1A*). In addition, we observed the midpiece length was significantly shorter in *Fbxo24* KO mouse spermatozoa compared with WT (21.67 ± 1.524 μm in WT vs. 14.80 ± 3.430 μm in *Fbxo24* KO) (*Figure 3E*). To better reveal spermatozoon abnormalities in *Fbxo24* KO mice, we decided to examine the ultrastructure of the spermatozoa using a scanning electron microscope (SEM). In WT spermatozoa, the midpiece was normally populated with many mitochondria in a spiraling fashion. However, in *Fbxo24* KO spermatozoa, mitochondria near the annulus were absent, resulting in a shorter mitochondrial sheath (*Figure 3F*). Transmission electron microscope (TEM) further confirmed that mitochondria were not present near the annulus or neck

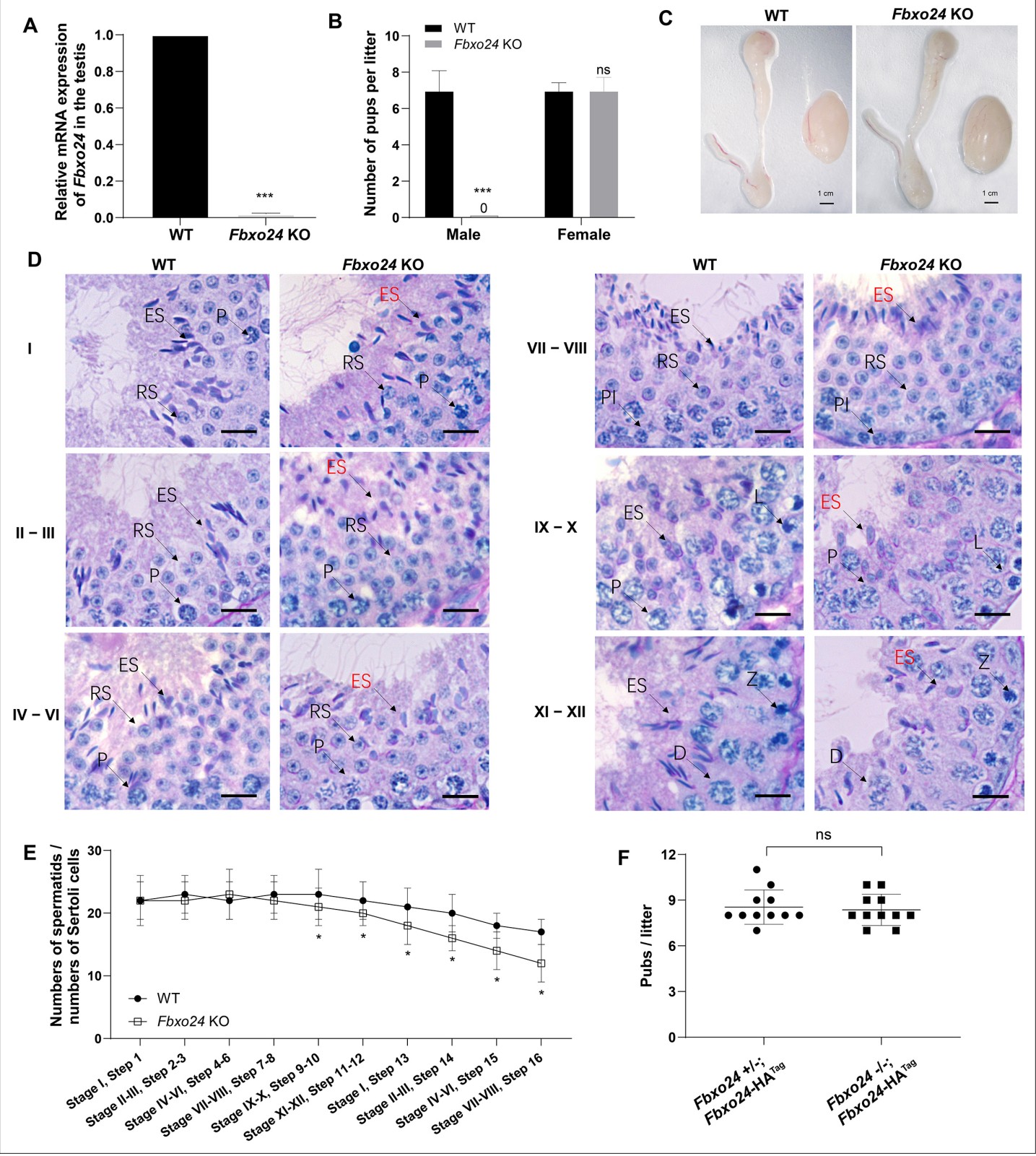

**Figure 2.** FBXO24 deletion impairs spermatogenic defects in the late steps of spermiogenesis. (**A**) qPCR analysis indicates that FBXO24 mRNA is markedly decreased in the testis of *Fbxo24* knockout (KO) as compared to wild-type (WT). *n* = 3 mice /group. ***p < 0.001. (**B**) The fertility tests of WT and *Fbxo24* KO male mice mated with fertile female mice are shown. *n* = 3 mice /group. Error bars represent mean ± standard deviation (SD). ***p < 0.001; ns: not significant. (**C**) Histological images of testes and epididymis of WT and *Fbxo24* KO mice at 8 weeks old are shown. (**D**) Periodic acid-Schiff

*Figure 2 continued on next page*

*Figure 2 continued*

(PAS)–hematoxylin staining of *Fbxo24* KO testis at 8 weeks old contained less- condensed late spermatids (red arrows). Spermatogenic stages are noted. RS, round spermatids; ES, elongating spermatids; Pl, preleptotene; P, pachytene; Z, zygotene; D, diplotene. Scale bars = 25 µm. (**E**) The number of late spermatids is significantly reduced in *Fbxo24* KO testis. Ratios of spermatids and Sertoli cells in tubule cross-sections of specific stages of seminiferous epithelial cycles and corresponding spermatid development steps are shown. *n* = 3 mice. Data are mean ± SD. *p < 0.05. (**F**) Litter sizes of mating tests. F1 generation of intercrosses between the indicated males and *Fbxo24*+/− females are shown. Each dot represents one litter.

The online version of this article includes the following source data and figure supplement(s) for figure 2:

**Figure supplement 1.** Spermiogenesis was defective in FBXO24-deficient mice.

**Figure supplement 1—source data 1.** Raw RT-PCR (Reverse transcriptase PCR) gel for *Figure 2—figure supplement 1B*.

and left apparent gaps along the mitochondrial sheath in *Fbxo24* KO spermatozoa (*Figure 3G*), indicating mitochondrial sheath formation was disrupted. In addition, some mitochondria in the midpiece of *Fbxo24* KO spermatozoa appeared large and vacuolar. To determine whether the failure in the assembly of the mitochondrial sheath could cause disrupted flagellar development, we examined cross-sections of the *Fbxo24* KO sperm flagellum. In the midpiece, well-defined outer dense fibers (ODFs) and the axoneme consisting of the typical '9 + 2' microtubules were observed in WT spermatozoa, whereas the *Fbxo24* KO spermatozoa displayed incomplete axonemes with missing axonemal microtubules (*Figure 3H*). In the principal piece, the *Fbxo24* KO spermatozoa also showed completely or partially lacking ODFs and a typical '9 + 2' microtubular structure (*Figure 3H*). These severe flagellar defects may explain why *Fbxo24* KO spermatozoa displayed decreased motility.

We next asked whether FBXO24 deletion influences the protein levels of mitochondrial and flagellum components in spermatozoa through Western blot analysis of spermatozoa. We found that the levels of MFN2 (a marker of mitochondrial fusion) and DRP1 (a marker of mitochondrial fission) were decreased in *Fbxo24* KO spermatozoa (*Figure 3I*; *Figure 3—figure supplement 1B*). Moreover, the protein expression of ODF2, AKAP3, and TSSK4, which were directly related to the formation of the sperm tail (*Wang et al., 2015*; *Donkor et al., 2004*; *Xu et al., 2020*), were all decreased in *Fbxo24* KO spermatozoa (*Figure 3J*; *Figure 3—figure supplement 1B*). This was supported by immunofluorescence analysis of ODF2 and AKAP3 with a reduced signals in *Fbxo24* KO flagella (*Figure 3—figure supplement 1C,D*). Interestingly, we found that the level of the CUL1, a scaffold for E3 ubiquitin ligase, was also significantly reduced in *Fbxo24* KO spermatozoa (*Figure 3J*). Together, these results indicate that FBXO24 is critical for mitochondrial sheath formation and could modulate the protein expression levels of mitochondrial and flagellar components to maintain sperm motility.

## The mitochondrial and CB architecture were affected in FBXO24-deficient round spermatids

To further explore whether the mitochondrial organelle morphology in round spermatids was affected upon FBXO24 depletion, we examined the ultrastructure of WT and *Fbxo24* KO testes by TEM. Notably, the mitochondria in the round spermatids of *Fbxo24* KO testis displayed vacuolar and had abnormal central cristae (*Figure 4A,B*). Interestingly, we identified that the CB was structurally maintained in *Fbxo24* KO round spermatids, whereas the size was often larger than WT (*Figure 4C,D*). CB was considered to have the functions in the assembly of the mitochondria during the spermiogenesis, and dysfunctional transformation of the CB in mouse spermatids could cause spermatozoa to possess a collapsed mitochondrial sheath (*Shang et al., 2010*), which resembled the phenotype observed in our *Fbxo24* KO mice. To explore the molecular reason for the abnormal CB formation, we examined the protein expression levels of CB components in WT and *Fbxo24* KO testes. Since MIWI was a core component of CB and also identified as an FBOX24 interacting partner from our immunoprecipitation-mass spectrometry (IP-MS) (*Supplementary file 1*), we focused on the examination of MIWI expression between WT and *Fbxo24* KO testes. Western blot analyses revealed that the protein level of MIWI was remarkably increased in the *Fbxo24* KO testes compared with WT (*Figure 4E*). However, the level of other proteins of CB, and PRMT6 (reported to be the substrate of FBXO24) (*Chen et al., 2020*) appeared to be unchanged in *Fbxo24* KO testes compared with WT testis (*Figure 4E*). These results suggest that FBXO24 is essential for maintaining the normal architecture of mitochondria and CB in the round spermatids.

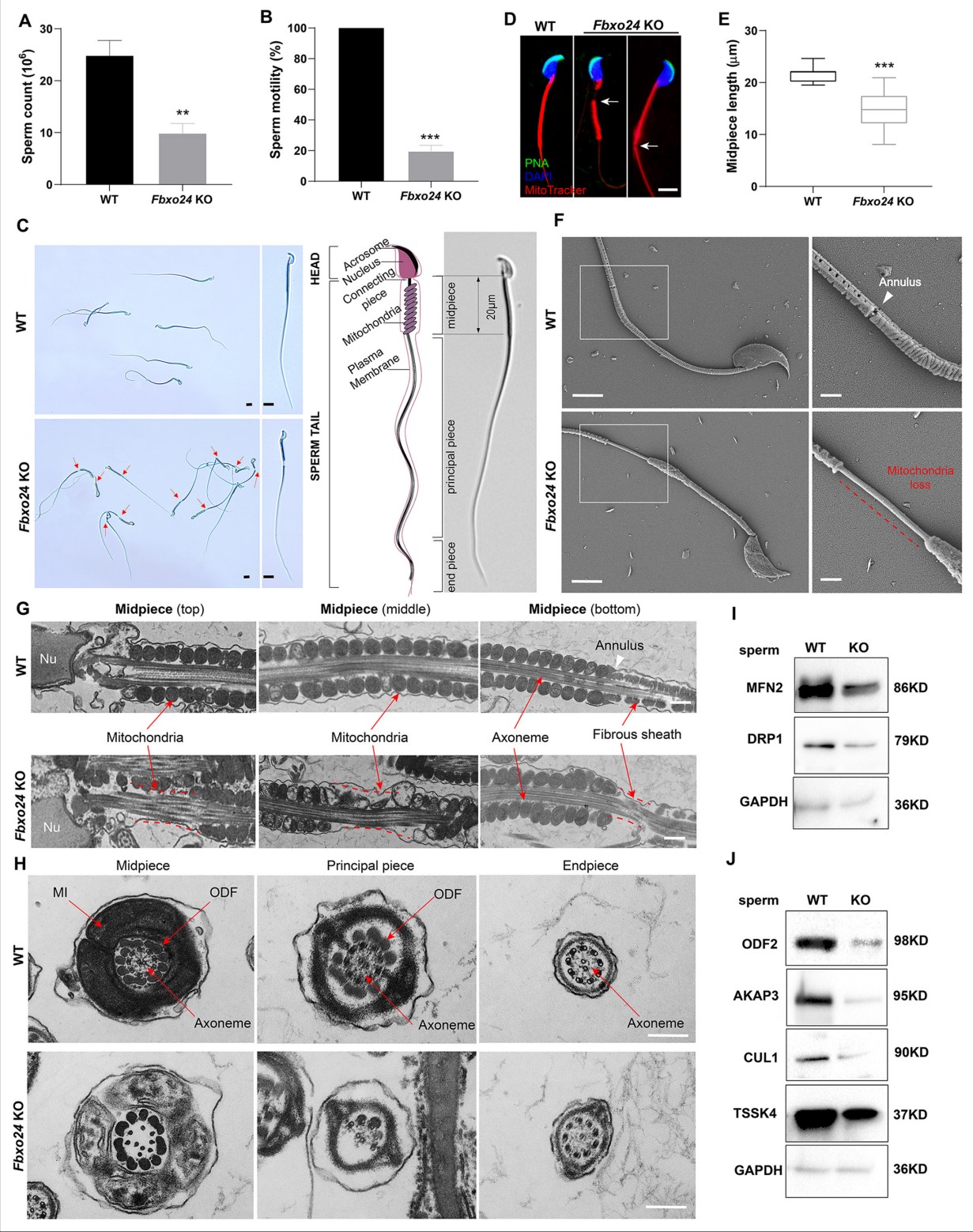

**Figure 3.** Sperm mitochondria and flagella are defective in FBXO24-deficient mice. Quantification of sperm counts (**A**) and sperm motility (**B**) from wild-type (WT) and *Fbxo24* knockout (KO) epididymis are shown. *n* = 3/group. Error bars represent mean ± standard deviation (SD). **p < 0.01. ***p < 0.001. (**C**) Sperm morphological images show the defective sperm of *Fbxo24* KO mice. Red arrows indicate abnormal gaps in the mitochondrial sheath. Scale bars = 5 μm. (**D**) Immunofluorescence images of sperm from WT and *Fbxo24* KO epididymis. PNA (acrosome, green), MitoTracker (mitochondria, red), and DAPI (nucleus, blue). White arrows indicate the weak or absent staining of MitoTracker. Scale bars = 5 μm. (**E**) Quantifications

*Figure 3 continued on next page*

*Figure 3 continued*

of the length of sperm midpiece from WT and *Fbxo24* KO mice are shown. *n* = 100/group. Error bars represent mean ± SD. ***p < 0.001. (**F**) Scanning electronic microscopy (SEM) images indicate the mitochondria detachment from the flagellum of *Fbxo24* KO sperm. Right panel insets show higher magnification of sperm midpiece. The arrowheads indicate the annulus. The dashed red line indicates a region where mitochondria are absent. Scale bars = 2 µm. (**G**) Transmission electronic microscopy (TEM) images indicate the mitochondria defects in three regions of *Fbxo24* KO sperm midpiece in the longitudinal sections. Nu, nucleus. The dashed red lines indicate a region where mitochondria are absent. The white arrowhead indicates sperm annulus. Scale bar = 0.2 µm. (**H**) TEM images indicate the ultrastructure of the midpiece, principal piece and end piece of WT and *Fbxo24* KO sperm flagellum in the cross-sections. Arrows indicate the mitochondria (MI), outer dense fiber (ODF) and axoneme. Scale bar = 0.2 µm. Western blot shows the levels of proteins of mitochondria (**I**) and axoneme (**J**) of WT and *Fbxo24* KO sperm. GAPDH serves as a loading control.

The online version of this article includes the following source data and figure supplement(s) for figure 3:

**Source data 1.** Raw western blot for *Figure 3I, J*.

**Figure supplement 1.** Sperm morphology analysis in FBXO24-deficient mice.

## Histones failed to be replaced in FBXO24-deficient mouse spermatozoa

To investigate the nuclear morphology of spermatozoa, we examined the sperm head structures of WT and *Fbxo24* KO spermatozoa by SEM. The *Fbxo24* KO spermatozoa exhibited abnormal head morphologies that deviated from the flat, crescent-shaped structures of their WT counterparts (*Figure 5A*). Specifically, the defects of *Fbxo24* KO spermatozoa included a disorganized anterior acrosome (AS) and the absence of a distinct equatorial segment (EQ), post-acrosomal sheath (PAS), ventral spur (VS), and sharp hook rim (HR) (*Figure 5A*). TEM analysis further revealed less-condensed nuclei of *Fbxo24* KO spermatozoa (*Figure 5B*), suggesting an abnormal nuclear chromatin compaction was affected in *Fbxo24* KO sperm head. *Fbxo24* KO spermatozoa exhibited elevated levels of DNA damage by TUNEL analysis (*Figure 5—figure supplement 1A*). In many KO mice studies, impaired chromatin condensation is frequently associated with abnormal sperm head morphology (*Okada, 2022*). We then investigated the potential causes of abnormal head structures in *Fbxo24* KO spermatozoa. Given that impaired histone-to-protamine exchange can result in less-condensed nuclei in spermatozoa, we examined histone and protamine levels in *Fbxo24* KO mouse sperm. Western blot revealed increased levels of all core histones H2A, H2B, H3, H4, and transition protein TNP1 but reduced levels of PRM2 in *Fbxo24* KO sperm (*Figure 5C,D*). We found that FBXO24 did not have the interactions with histones H2A, H2B, H3, H4, and transition protein TNP1 (*Figure 5—figure supplement 1B*). Moreover, we found a reduced protein level of PHF7, TSSK6, and RNF8 in *Fbxo24* KO testis (*Figure 5E,F*), which have been reported to regulate the chromatin structure to facilitate histone-to-protamine replacement (*Jha et al., 2017*; *Lu et al., 2010*; *Wang et al., 2019*). These data suggest that FBXO24 deletion causes an incomplete histone-to-protamine exchange and defective chromatin compaction during spermiogenesis.

## FBXO24 depletion leads to aberrant expression of spermiogenic genes in the round spermatids

To further investigate the underlying molecular changes in sperm formation upon FBXO24 depletion, we performed RNA-seq using purified round spermatids (RS) from adult *Fbxo24* KO and WT males (*Figure 6—figure supplement 1A*). RNA-seq analysis showed a large number of genes significantly differentially expressed in *Fbxo24* KO round spermatids, including 5368 up-regulated and 4097 down-regulated genes (*Figure 6A* and *Supplementary file 2*). The Gene Ontology (GO) analysis revealed that the down-regulated genes were related to mitochondrion localization, chromatin organization, and protein K48-linked ubiquitination (*Figure 6B* and *Supplementary file 3*). The expression of many critical genes involved in mitochondrion localization (e.g., *Hif1a*, *Nefl*, *Mgarp*, *Zmynd12*, *Map7d1*, *Bbs7*, *Mfn1*, and *Ube2j2*) and chromatin organization (e.g., *Rnf20*, *Ep300*, *Sirt1*, *Hmgb2*, *Hsf1*, *Cul4b*, *Kat5*, and *Phf7*) were significantly decreased in *Fbxo24* KO round spermatids (*Figure 6C,D*). Interestingly, we found that *Sept4* and *Sept12*, the members of *Septin* family known to generate annuli of spermatozoa, were also down-regulated in *Fbxo24* KO round spermatids (*Figure 6—figure supplement 1B*). The most up-regulated genes in *Fbxo24* KO round spermatids were mainly related to cellular metabolic processes and organelle organization (*Figure 6—figure supplement 1C*). Furthermore, we identified an overlap between the down-regulated genes and the altered alternative splicing (AS) genes in *Fbxo24* KO round spermatids (*Figure 6E*). A large number of mRNA splicing events

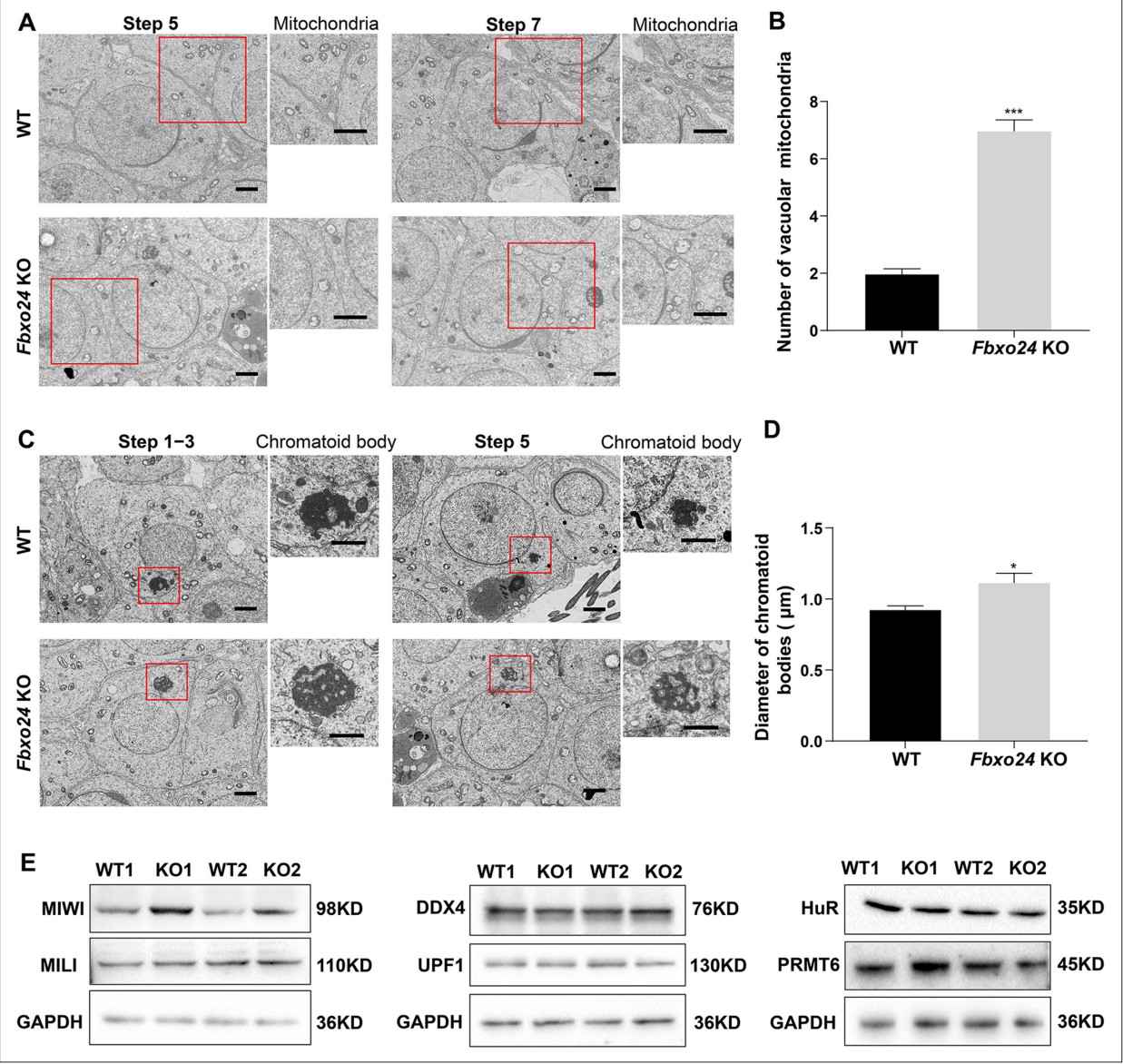

**Figure 4.** Ablation of FBXO24 affects mitochondria and chromatoid body (CB) architecture in the round spermatids. (**A**) Transmission electron microscope (TEM) images show vacuolar mitochondria with disorganized cristae in the round spermatids of *Fbxo24* knockout (KO) testes. Right panel insets show higher magnification of mitochondria. Scale bars = 1 µm. (**B**) Quantification of the number of vacuolar mitochondria. Error bars represent mean ± SD. n = 3. \*\*\*p < 0.001. (**C**) TEM images showing decondensed and enlarged CB with an irregular network in the round spermatids of *Fbxo24* KO testes. Right panel insets show a higher magnification of CB. Scale bars = 1 µm. (**D**) Quantification of size/diameters of CB. Error bars represent mean ± SD. n = 3. \*p < 0.05. (**E**) Western blot analysis expression levels of CB components and PRMT6 in testes from wild-type (WT) and *Fbxo24* KO mice at 8 weeks old. GAPDH serves as a loading control.

The online version of this article includes the following source data for figure 4:

**Source data 1.** Raw western blot for *Figure 4E*.

were identified in *Fbxo24* KO round spermatids (*Figure 6F*), including skipped exons (SE), alternative 5′ splice sites (A5SS), alternative 3′ splice sites (A3SS), mutually exclusive exons (MXE), and retained introns (RI). Interestingly, GO analysis of the abnormally spliced genes highlighted the functional categories involved in sperm formation, such as outer dynein arm assembly, mitochondrion localization, and chromatin organization (*Figure 6—figure supplement 1D*). Thus, the RNA-seq data revealed genetic ablation of *Fbxo24* has a significant impact on the transcriptome of round spermatids, indicating that FBXO24 is required for gene expression in round spermatids during spermiogenesis.

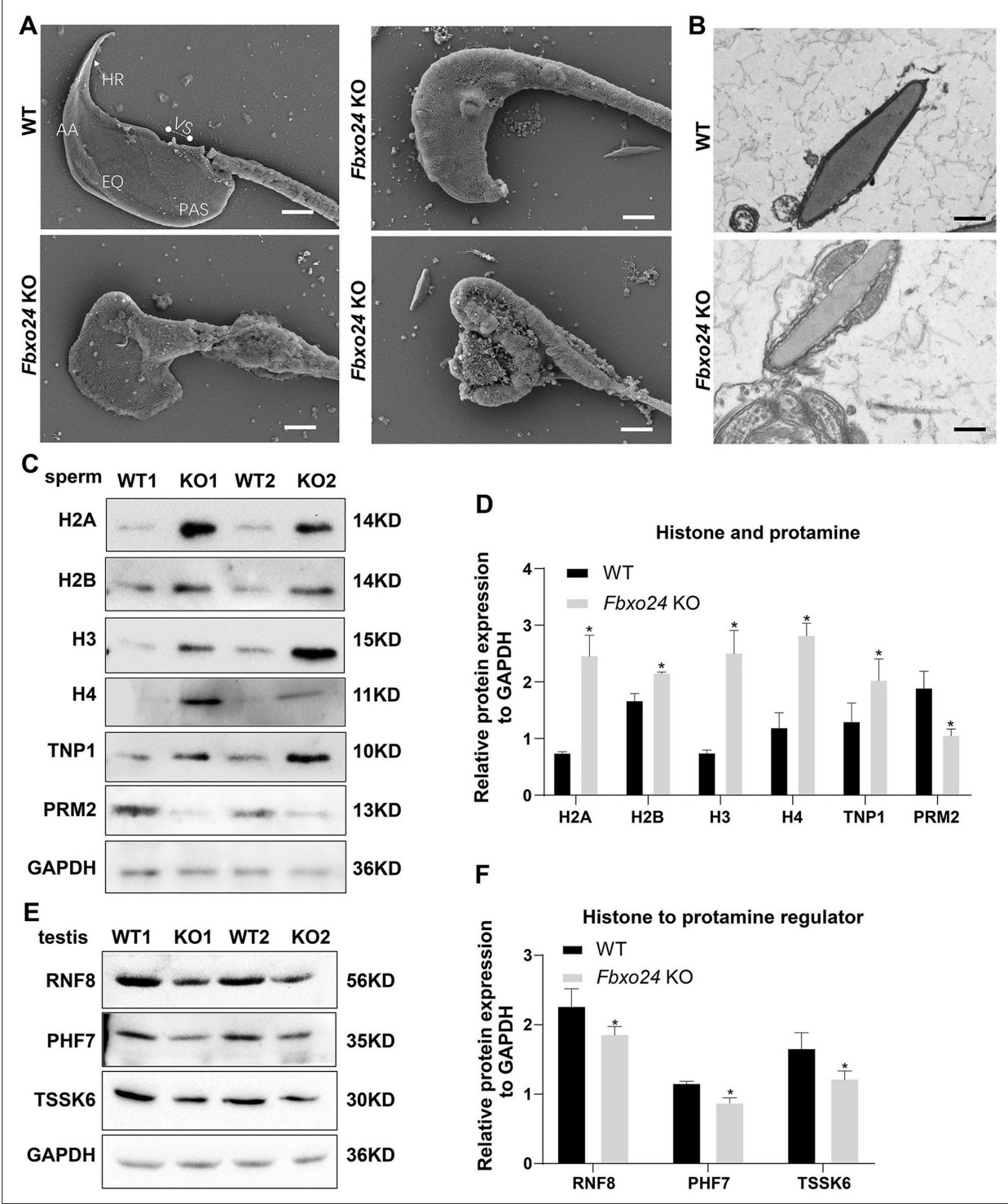

**Figure 5.** FBXO24 deficiency in mice impairs sperm histone-to-protamine exchange. (**A**) Scanning electron microscope (SEM) images show the abnormality of *Fbxo24* knockout (KO) sperm head. AA, anterior acrosome; EQ, equatorial segment; PAS, post-acrosomal segment; VS, ventral spur; HR, hook rim. Scale bars = 2 μm. (**B**) Transmission electron microscopy (TEM) images show the decondensed nucleus (Nu) of *Fbxo24* KO sperm. Scale bars = 1 μm. (**C**) Western blot analysis of the expression of histones (H2A, H2B, H3, and H4), transition proteins (TNP1), and protamines (PRM2) from wild-type (WT) and *Fbxo24* KO sperm are shown. GAPDH serves as a loading control. (**D**) Quantification of protein levels. Error bars represent mean ± standard deviation (SD), *n* = 3. *p < 0.05. (**E**) Western blot analysis of the expression of indicated proteins in WT and *Fbxo24* KO testis. GAPDH serves as a loading control. (**F**) Quantification of protein levels. Error bars represent mean ± SD, *n* = 3. *p < 0.05.

The online version of this article includes the following source data and figure supplement(s) for figure 5:

*Figure 5 continued on next page*

*Figure 5 continued*

**Source data 1.** Raw western blot for *Figure 5C, E*.

**Figure supplement 1.** DNA damage, histone, and transition protein analysis.

**Figure supplement 1—source data 1.** Raw western blot for *Figure 5—figure supplement 1B*.

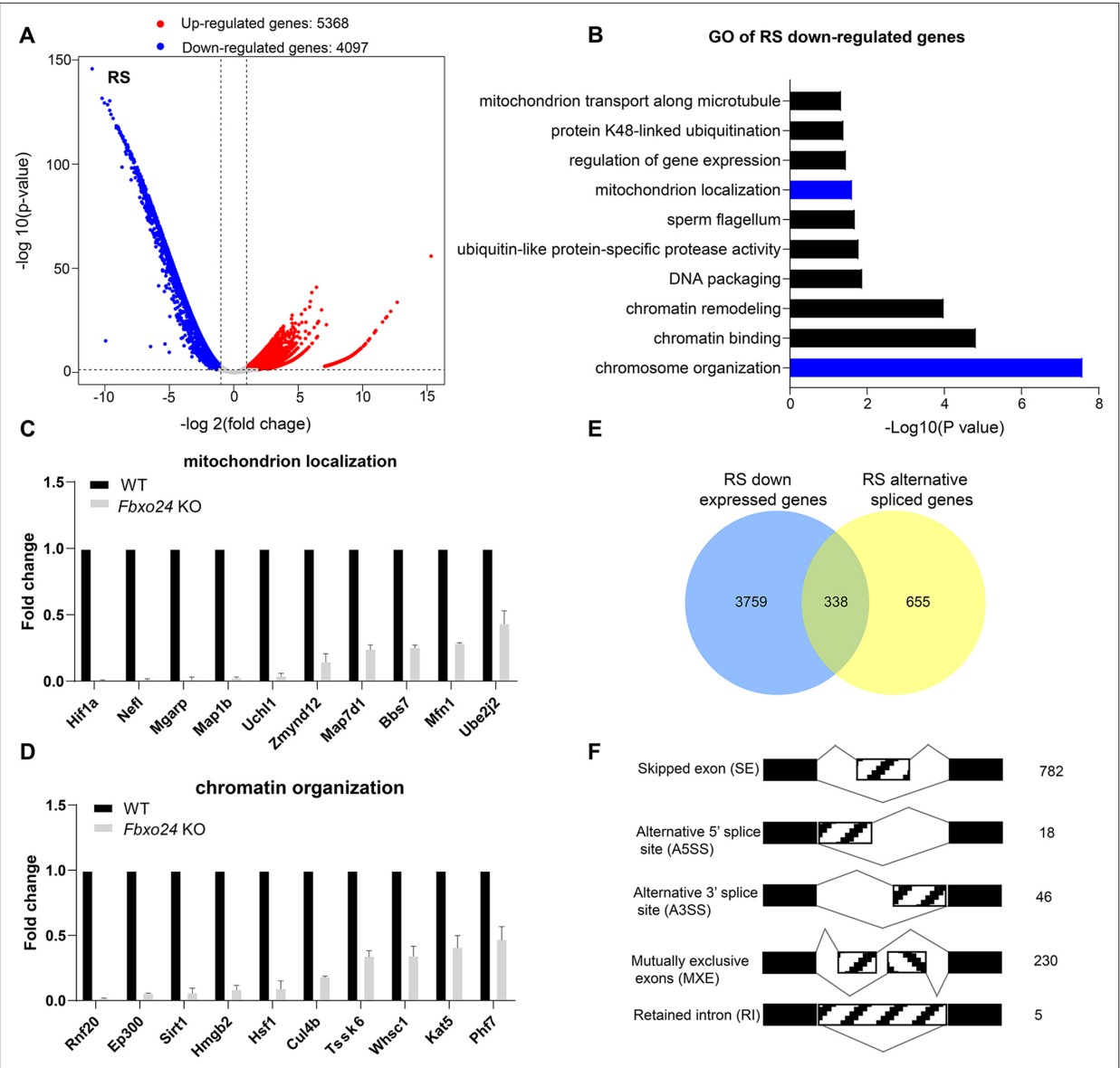

**Figure 6.** RNA-seq analyses of the round spermatids from FBXO24-deficient testes. (**A**) Volcano plot of differentially expressed transcripts in the round spermatids (RS) of *Fbxo24* knockout (KO) vs. wild-type (WT) mice. Each red (up-regulation) or blue (down-regulation) dot represents a significantly changed gene. (**B**) Gene Ontology (GO) term enrichment analysis of down-regulated transcripts of *Fbxo24* KO RS. Gene expressions of mitochondrion localization (**C**) and chromatin organization (**D**) in RNA-seq analysis. (**E**) Venn diagrams showing the overlap between down-regulated genes and abnormal alternative splicing genes in *Fbxo24* KO RS. (**F**) Summary of differential splicing evens in *Fbxo24* KO RS. The number of each category of alternative splicing is indicated.

The online version of this article includes the following figure supplement(s) for figure 6:

**Figure supplement 1.** Global gene expression altered in round spermatids of FBXO24-deficient mice.

## Loss of FBXO24 results in aberrant mRNA alternative splicing in the round spermatids

Since FBXO24 expressed in the nucleus of the round spermatids, we speculated that FBXO24 might interact with the key splicing factors. To test this hypothesis, we performed IP-MS using the HA antibody to unbiasedly identify the interactome of FBXO24 in testes. The IP-MS results showed many splicing factors, such as SRSF2, SRSF3, and SRSF9, highly enriched in HA-immunoprecipitates (*Supplementary file 1*). Through co-immunoprecipitation assays, we confirmed FBXO24 has indeed interacted with splicing factors SRSF2, SRSF3, and SRSF9 in the testes (*Figure 7A*). Interestingly, we found a significant decrease in SRSF2, SRSF3, and SRSF9 protein expression levels in *Fbxo24* KO testes compared to controls (*Figure 7B,C*). Because many alternative splicing genes related to sperm formation were affected in *Fbxo24* KO round spermatids by RNA-seq analysis, we asked if the splicing levels of corresponding exons of those genes were altered. To this end, we performed RT-PCR on purified round spermatids to verify the alternative splicing alteration. The RT-PCR results showed that many genes related to sperm formation, including mitochondria (*Mfn1*), flagellum (*Zmynd12*, *Map7d1*, and *Bbs7*), chromatin (*Phf7* and *Kat5*), acrosome (*Nucb2*), appeared to be significantly alternative splicing changes in their corresponding exons (*Figure 7D–G*). In addition, we found that the alternative splicing of *Ube2j2*, a ubiquitin-conjugating enzyme, was also affected in *Fbxo24* KO round spermatids (*Figure 7H*). Of note, sperm motility of *Ube2j1* KO mice was also severely impaired (*Koenig et al., 2014*). Together, these results indicate that FBXO24 could interplay with key splicing factors to regulate the mRNA splicing of some essential genes related to sperm formation.

## FBXO24 mediates K48-linked polyubiquitination of MIWI

Since the increased expression of MIWI was found in the testes of *Fbxo24* KO mice, we sought to determine whether FBXO24 interacts with MIWI and affects its degradation via ubiquitination. We ectopically expressed FBXO24-mCherry in HEK293T cells and incubated FBXO24-mCherry with the testicular protein lysate. MIWI and CUL1 were detected in FBXO24 immunoprecipitates (*Figure 8A*), suggesting that FBXO24 could bind MIWI and CUL1 in vivo. To confirm that FBXO24 endogenously interacted with its candidates, we performed immunoprecipitation (IP) with the testis of *Fbxo24*-HA-tagged transgenic mice. Through co-immunoprecipitation assays, we identified that FBXO24 indeed interacts with MIWI and SCF subunits, including CUL1, SKP1, and RBX1. (*Figure 8B*). To further dissect the binding abilities between FBXO24 and MIWI, we examined which domain was responsible for the interaction through the truncated proteins of FBXO24, which only contain the F-box or RCC1 domain (*Figure 8C*). The IP results showed that both the F-box and RCC1 domains of FBXO24 could bind with MIWI (*Figure 8D*). Furthermore, we found that FBXO24 leads to a decreased level of MIWI in a concentration-dependent manner, suggesting FBXO24 modulates the degradation of MIWI (*Figure 8E*). Because K48- and/or K63-linked polyubiquitination was reported to be responsible for degradation, trafficking, and phosphatase activation (*Tracz and Bialek, 2021*), we then asked which kind of modification participates in MIWI ubiquitination. We found that overexpression of FBXO24 enhanced polyubiquitination of MIWI in the presence of Ub (K48) but did not enhance polyubiquitination of MIWI in the presence of Ub (K63) (*Figure 8F*). Interestingly, we also found that FBXO24 contributes to the decreased expression of MIWI in a ubiquitin-dependent manner, and the ubiquitin level of MIWI appeared to be reduced in FBXO24 KO round spermatids (*Figure 8G,H*). These data suggest that FBXO24 interacts with SCF subunits and mediates the degradation of MIWI via K48-linked polyubiquitination.

## FBXO24 is required for normal piRNA production in the testes

Because piRNAs could be loaded onto MIWI (*Kawase et al., 2022*), we next examined whether the population of piRNAs was affected in *Fbxo24* KO testes. Sequencing of small RNA libraries constructed from total RNA revealed that the expression of miRNAs was not extensively changed in *Fbxo24* KO testes, with only 20 up- and 13 down-regulated miRNAs (*Figure 9A*; *Figure 9—figure supplement 1A*, and *Supplementary file 4*). However, a large number of piRNAs were identified to be differentially expressed in *Fbxo24* KO testes, 463 up- and 128 down-regulated piRNAs (*Figure 9—figure supplement 1B* and *Supplementary file 5*). After normalized with miRNA counts, the relative amount of total piRNAs was also increased in *Fbxo24* KO testes (*Figure 9—figure supplement 1C*). Specifically, an increased piRNAs profile was observed in the genomic regions between 2 kb upstream

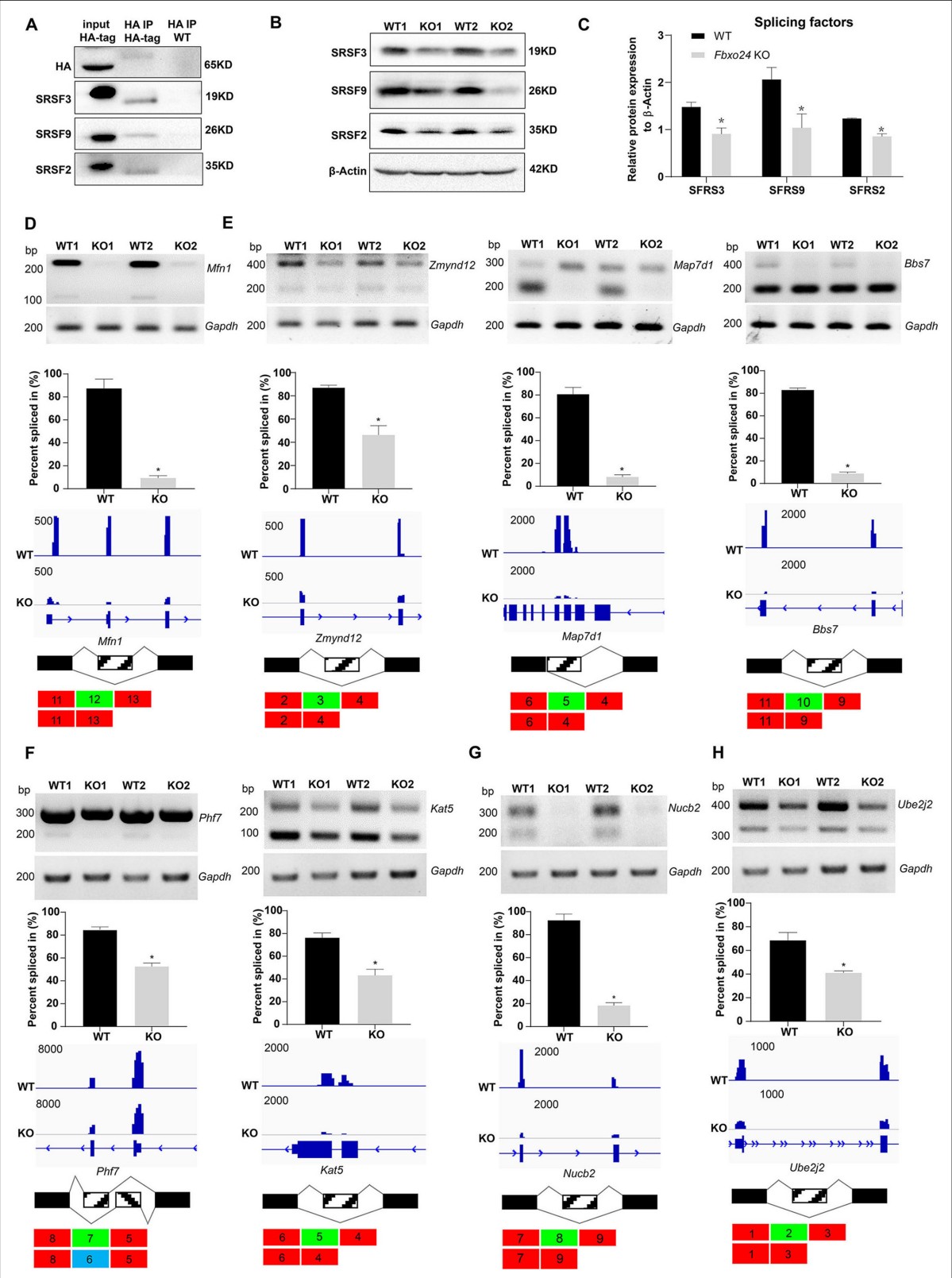

**Figure 7.** Aberrant alternative splicing of spermiogenesis genes in the round spermatids of FBXO24-deficient mice. (**A**) Co-immunoprecipitation analysis of FBXO24 and the splicing regulators (SRSF2, SRSF3, and SRSF9) in *Fbxo24*-HA-tagged mice testis. Wild-type (WT) testis was used as a negative control. (**B**) Western blotting analysis of the splicing regulators in WT and *Fbxo24* knockout (KO) testis. GAPDH was used as a loading control. (**C**) Quantification of protein levels. Error bars represent mean ± standard deviation (SD), *n* = 3. *p < 0.05. Validation of abnormal alternative splicing

*Figure 7 continued on next page*

*Figure 7 continued*

genes related to (**D**) mitochondria (*Mfn1*), (**E**) flagellum (*Zmynd12*, *Map7d1*, and *Bbs7*), (**F**) chromatin (*Phf7* and *Kat5*), (**G**) acrosome (*Nucb2*), and (**H**) ubiquitination (*Ube2j2*). The top panels represent RT-PCR analysis of indicated genes in WT and *Fbxo24* RS. *Gapdh* serves as a loading control. The middle panels show the quantification of percent spliced in (PSI) and alternative sites in RNA-seq. Error bars represent mean ± SD, *n* = 2. *p < 0.05. The bottom panels represent the schematic diagram of alternative sites exons.

The online version of this article includes the following source data for figure 7:

**Source data 1.** Raw western blot for *Figure 7A, B*.

**Source data 2.** Raw RT-PCR gel for *Figure 7D–H*.

of transcriptional start sites (TSS) and 2 kb downstream of transcriptional end sites (TES) in *Fbxo24* KO testes (*Figure 9B*). To assess the potential functions of the association of piRNAs to different functional gene regions, we analyzed the piRNAs mapping density of varying gene regions, including coding region (CDS), 5′ and 3′ untranslated region (UTR), and intron, as well as transposable element (TE), including retrotransposon (LTR, LINE, and SINE) and DNA transposon. The results showed that the intron- and retrotransposon-derived piRNAs were the most affected types in *Fbxo24* KO testes (*Figure 9C* and *Supplementary file 6*). Furthermore, we analyzed the abundance and size of piRNA populations in adult *Fbxo24* KO and WT testes. The results showed that the number of 29–31 nt piRNAs were remarkably increased in *Fbxo24* KO testes compared with WT (*Figure 9D*). In addition, we found many up-regulated piRNAs in *Fbxo24* KO testes were also presented in previously published data of MIWI immunoprecipitates (*Reuter et al., 2011*; *Figure 9E*), suggesting that the increased piRNAs were MIWI-bound piRNAs. The first nucleotide and 10th nucleotide of repeat piRNAs of *Fbxo24* KO testes exhibited a similar strong bias compared with piRNAs of WT testes, indicating the amplification process of the ping-pong cycle was not affected (*Figure 9—figure supplement 1D*). Together, these results show that the loss function of FBXO24 results in aberrant piRNA production in testes, suggesting FBXO24 is related to normal piRNA counts of the testis.

## Discussion

In this study, we identified FBXO24 as a testis-enriched F-box protein conserved in mammals and highly expressed in the round spermatids and elongating spermatids. To define its biological roles in vivo, we generated genetically engineered *Fbxo24* KO and *Fbxo24* HA-tagged transgenic mouse models. Our functional study demonstrated that FBXO24 is essential for spermiogenesis and piRNA production. FBXO24 deficiency leads to remarkable histone retention and the disruption of histone-to-protamine transition in the mature sperm. A growing number of studies have demonstrated impaired chromatin condensation can cause nuclear damages as DNA denaturation or fragmentation often associated with male infertility (*Filatov et al., 1999*). The molecular basis of nuclear condensation is related to key events such as the expression of testis-specific histone variants, post-translational modifications of histones, and transient DNA strand breaks. Indeed, we found that, in the current study, FBXO24 could modulate the homeostasis of the histone-to-protamine transition regulators PHF7 (*Wang et al., 2019*), TSSK6 (*Jha et al., 2017*), and RNF8 (*Lu et al., 2010*), while RNF8 was reported to be independent to the process (*Abe et al., 2021*).

Although the abnormalities of mitochondrial sheaths are observed in infertile men (*Kubo-Irie et al., 2005*), the key proteins implicated in sperm mitochondrial sheath formation largely remain unclear due to a lack of good animal models with the typical phenotype. Notably, we found that the round spermatids and spermatozoon of *Fbxo24* KO mice have impaired mitochondrial architecture, with mitochondrial size heterogeneity and a more open structure of the mitochondria with less crista. The defective mitochondria were associated with energetic disturbances and reduced sperm motility. Using MitoTracker staining and an electron microscope, we demonstrated the mitochondrial structure is severely disrupted in *Fbxo24* KO spermatozoa. Previous studies showed abnormalities of the annulus (*Sept4* and *Sept12*) (*Kissel et al., 2005*; *Shen et al., 2017*), and chromatid bodies (*Tssk1* and *Tssk2*) (*Shang et al., 2010*; *Dirami et al., 2015*) can disrupt abnormal mitochondrial coiling. The CB was considered to migrate to the caudal end of the developing middle piece of the flagellum, moving in front of the mitochondria that are engaged in mitochondrial sheath morphogenesis (*Fawcett et al., 1970*). This study revealed that FBXO24 regulates spatiotemporal mitochondrial dynamics during spermiogenesis. FBXO24 might serve as a critical protein in regulating sperm mitochondrial sheath

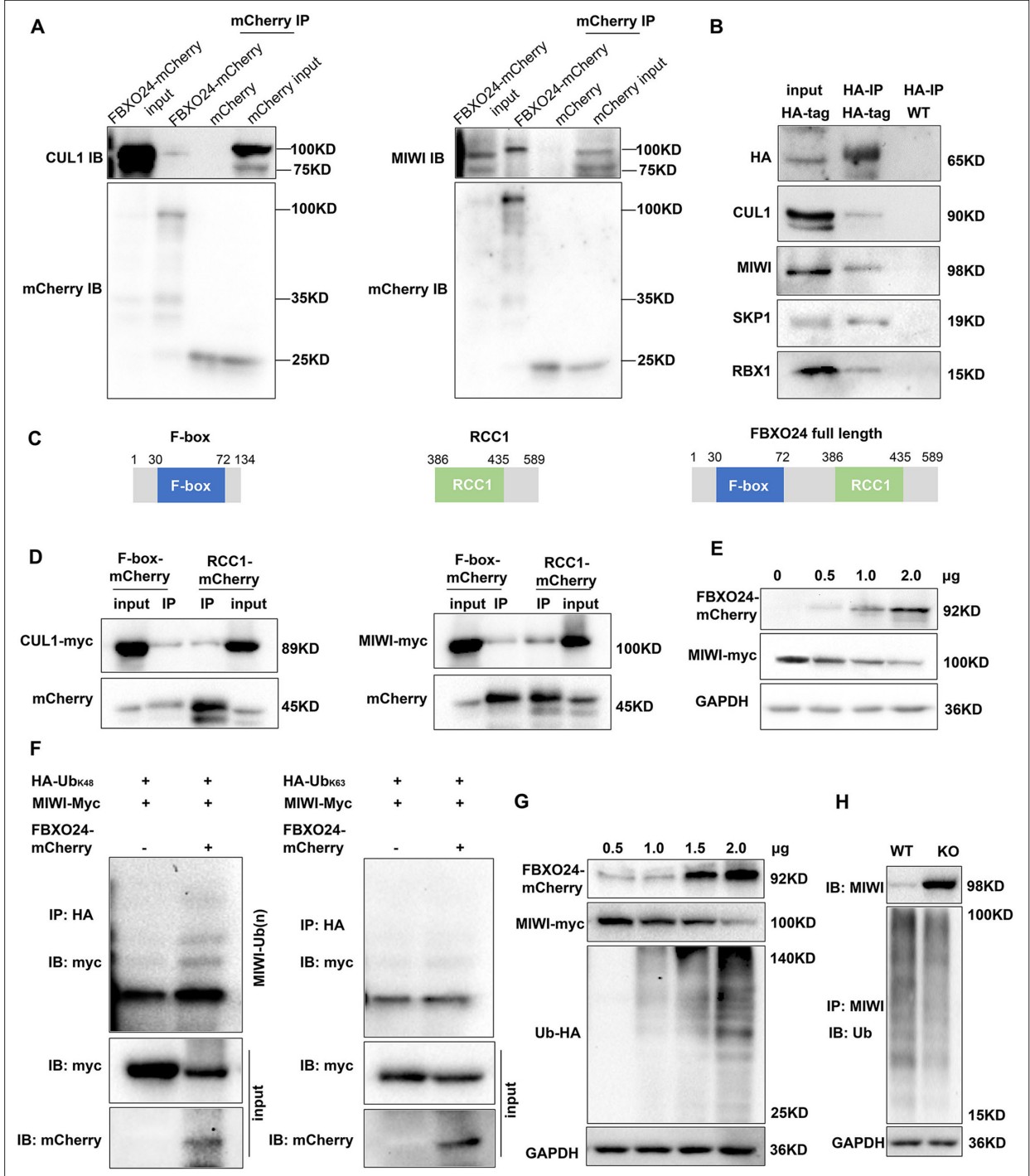

**Figure 8.** FBXO24 interacts with MIWI and mediates its K48-linked polyubiquitination. (**A**) HEK293T cells transfected with empty FBXO24-mCherry or mCherry vector. Anti-mCherry beads were used for immunoprecipitation (IP), and western blots were used to detect the CUL1 (left panel) and MIWI (right panel) expression. (**B**) The testis lysate of *Fbxo24*-HA-tagged mice were immunoprecipitated with anti-HA beads. Western blots were used to detect the HA, CUL1, MIWI, SKP1, and RBX1expression. (**C**) Schematic structures of the truncated FBXO24 protein are shown. Broken boxes show the domain of F-box and regulator of chromosome condensation 1 (RCC1). (**D**) HEK293T cells were transfected with indicated plasmids. IP was performed using the anti-mCherry antibody. (**E**) Western blot analysis of HEK293T cells transfected with indicated FBXO24-mCherry and 2 µg MIWI-myc plasmids. The cell lysates were immunoblotted with anti-mCherry and anti-myc antibodies. (**F**) FBXO24 mediated the ubiquitination of MIWI in the presence of Ub (K48) not Ub (K63). (**G**) HEK293T cells were transfected with indicated FBXO24-mCherry, 2 µg MIWI-myc, and 2 µg Ub-HA plasmids. The cell lysates were immunoblotted with the anti-mCherry, anti-myc, and anti-HA antibodies. GAPDH serves as a loading control. (**H**) Ubiquitination analysis of MIWI in the round spermatids of Fbxo24 knockout (KO) mice. The cells were treated with MG132 (10 µM) in the ubiquitination assay.

*Figure 8 continued on next page*

*Figure 8 continued*

The online version of this article includes the following source data for figure 8:

**Source data 1.** Raw western blot for *Figure 8A, B, D–H*.

formation, which expands the knowledge of sperm mitochondria formation. In addition, we found that FBXO24 is indispensable for proper flagellum formation, such as axoneme and ODFs. Therefore, the reduction of sperm motility of *Fbxo24* KO male mice observed in this study might attribute to the uncompleted mitochondrial formation of the middle piece and defective flagellum assembling.

It is worth mentioning that the RNA-seq data of the round spermatids showed that FBXO24 ablation leads to dysregulation in the mRNA expression of many genes related to chromatin, mitochondria, and flagellum. We found mRNA expression changes were related to the mRNA alternative splicing in *Fbxo24* KO round spermatids. Combined with our IP evidence of molecular interactions between FBXO24 and the key splice factors (SRSF2, SRSF3, and SRSF9), it is reasonable to infer that dysregulation of these splicing factors inevitably would lead to more splicing errors in their target genes, thus amplifying the initial adverse effects and generating a vicious circle of aberrant splicing. Of note, many genetic changes identified in this study have been closely associated with sperm formation. For example, *Zmynd12* (*Dacheux et al., 2023*), *Bbs7* (*Zhang et al., 2013*), and *Ube2j2* (*Koenig et al., 2014*) were required for flagellum function and male fertility. *Map7d1* facilitates microtubule stabilization (*Kikuchi et al., 2022*) and *Mfn1* deficiency leads to defects in mitochondrial activity and male infertility (*Zhang et al., 2016*). *Phf7* modulates BRDT stability and histone-to-protamine exchange during spermiogenesis (*Kim et al., 2020*). *Kat5* encodes an essential lysine acetyltransferase, which is involved in regulating gene expression and chromatin remodeling (*Gehlen-Breitbach et al., 2023*). *Nucb2* suppresses the acrosome reaction in sperm within the mouse epididymis (*Kim et al., 2023*). Therefore, our study, for the first time, provides evidence that FBXO24 interacts with the splicing factors to regulate the alternative splicing of mRNA involved in round spermatid development. However, the exact regulatory mechanism of how FBXO24 facilitates the mRNA alternative splicing needs further study.

MIWI and piRNAs are highly abundant in round spermatids, but their levels start to decline normally in elongating spermatids and are completely eliminated in matured sperm (*Deng and Lin, 2002*). Interestingly, Mo-Fang Liu et al. reported that piRNA-triggered MIWI degradation occurs in late spermatids through the APC/C-26S proteasome pathway in vitro, which in turn leads to piRNA elimination (*Zhao et al., 2013*). As many piRNAs also appear to have the capacity to target diverse mRNA, they demonstrated that piRNAs act extensively as siRNAs to degrade specific mRNA (*Zuo et al., 2016*; *Gou et al., 2014*). Overexpression of a piRNA cluster in the mouse genome can reduce the expression of mRNA required for spermatogenesis, leading to male infertility (*Goh et al., 2015*). The level of 29–31 nt piRNAs was remarkably increased in *Fbxo24* KO testes, which raises a possibility that piRNA-related transcriptional repression might contribute to the reduction of mature sperm in *Fbxo24* KO mice. In the current study, we identified that testis-enriched FBXO24 could interact with MIWI and mediate its degradation via the K48-linked ubiquitination, which provides new insight into MIWI degradation during spermiogenesis.

In summary, this study revealed a novel role for FBXO24 in controlling mRNA alternative splicing of round spermatids and MIWI/piRNA pathway, and demonstrated that FBXO24 is required for mitochondrial organization and chromatin condensation in mouse spermatozoa (*Figure 10*). The results disclosed FBXO24 interacts with splicing factors to regulate the mRNA splicing of some essential genes required for spermiogenesis, which probably functionally connects the dysregulation of genes involved in mitochondria migration and histone removal during spermiogenesis. FBXO24 interplays with the subunits of the SCF complex and mediated MIWI degradation. This study extensively expands our understanding of the mechanistic explanation of FBXO24 for the regulation of spermiogenesis.

# Materials and methods
## Ethics statement

All the animal procedures were approved by the Institutional Animal Care and Use Committee protocols (#S2795) of Tongji Medical College, Huazhong University of Science and Technology, and the

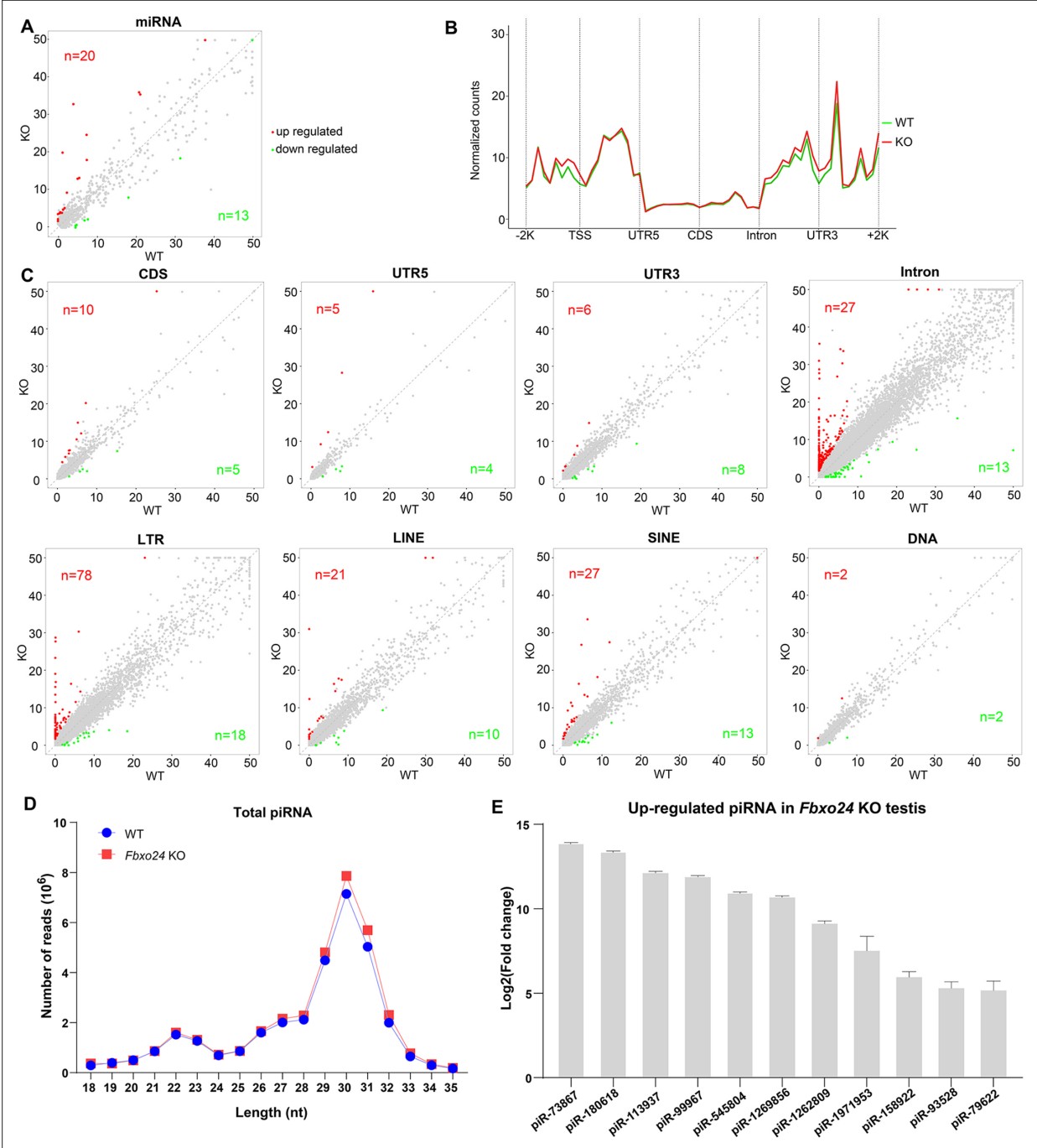

**Figure 9.** Small RNA-seq analysis of testes from FBXO24-deficient mice. (**A**) A scatter plot of differentially expressed miRNA is shown. Red and green dots represent up- and down-regulated miRNA (fold change >2, p < 0.05), respectively. (**B**) Genomic distribution of piRNA profile in *Fbxo24* knockout (KO) vs. wild-type (WT) testis. piRNA levels were examined in each 200-bp interval of a 2-kb region up- and downstream of the annotated genes. (**C**) Scatter plots of differentially expressed piRNA mapping density (reads/kb) of the coding region (CDS), 5′ and 3′ untranslated region (UTR), and intron, as well as transposable element (TE), including retrotransposon (LTR, LINE, and SINE) and DNA transposon. The piRNA read counts were normalized with miRNA. Red and green dots represent up- and down-regulated piRNA (fold change >2, p < 0.05), respectively. (**D**) The size distribution of piRNAs in *Fbxo24* KO vs. WT testis. (**E**) The top 10 up-regulated piRNAs in *Fbxo24* KO testis exist in MIWI immunoprecipitates of GSM822760 data.

The online version of this article includes the following figure supplement(s) for figure 9:

**Figure supplement 1.** miRNA and piRNA expression analysis in the FBXO24-deficient mice.

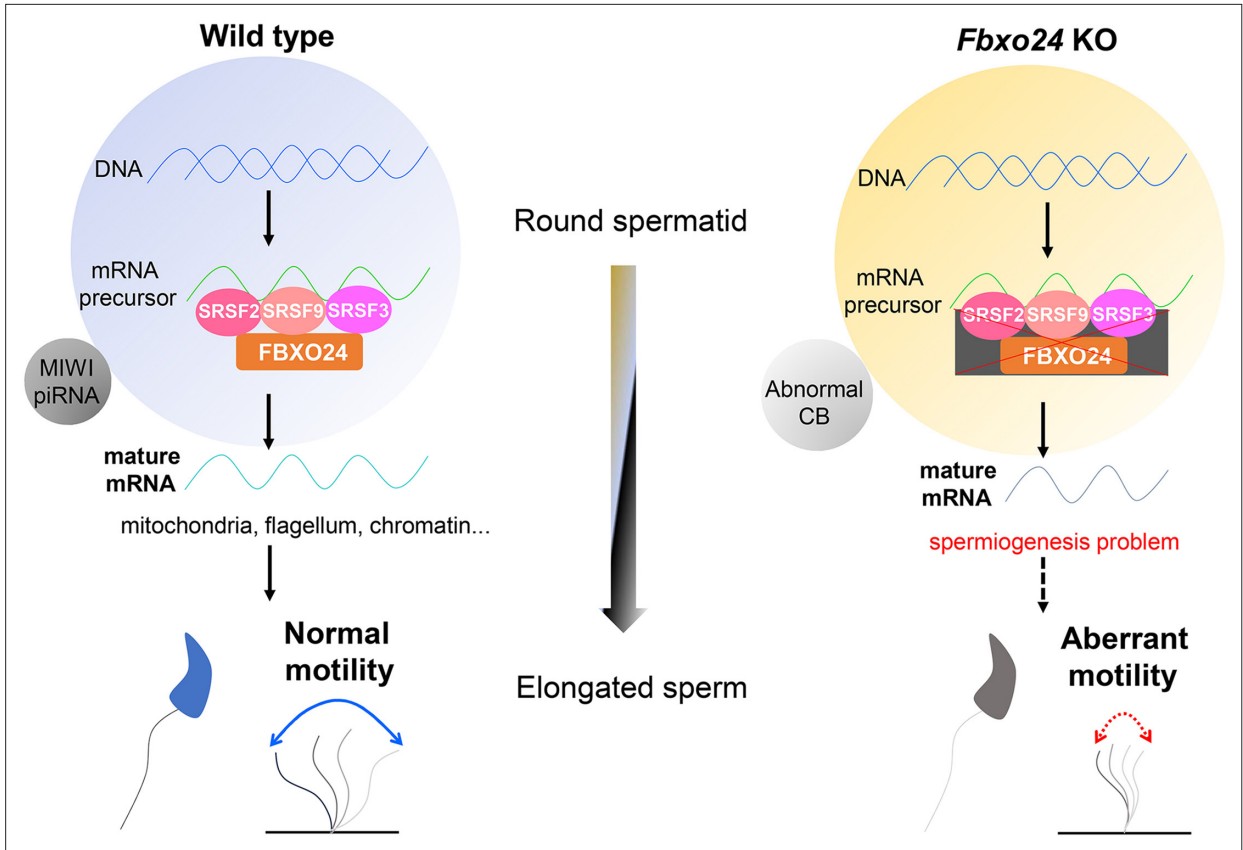

**Figure 10.** A schematic model shows the FBXO24-mediated post-transcriptional regulation during spermiogenesis. Fbxo24 interacts with key splicing factors (SRSF2, SRSF3, and SRSF9) to coordinate proper alternative splicing of the target mRNA transcripts involved in spermiogenesis. FBXO24 regulates the architectures of mitochondria and chromatid body through MIWI/piRNA pathway in the round spermatids.

mice were housed in the specific pathogen-free facility of Huazhong University of Science and Technology. All experiments with mice were conducted ethically according to the Guide for the Care and Use of Laboratory Animal guidelines.

### Generation of *Fbxo24* mutant mice

*Fbxo24* KO mice were generated by CRISPR/Cas9 technology. Briefly, the two pairs of single-guided RNAs (sgRNAs) with the sequence sgRNA-1: 5′-TGTGGAGGCGCATCTGTCGAAGG-3′ and sgRNA-2 5′-GTCAAAGACTTGGTCGCCCTAGG-3′ were designed for targeting for the third exon of *Fbxo24* gene. The sgRNAs were injected into the pronuclei of fertilized eggs, and the two-cell stage embryos were transferred into the oviducts of pseudopregnant C57BL/6J females in the next day to generate *Fbxo24* mutant founder mice (F0). The F1 *Fbxo24* heterozygous mice were produced by crossing the F0 founder mice with WT mice and further intercrossed F1 to obtain *Fbxo24* KO mice carrying an 82-bp deletion in exon 3 of the *Fbxo24* gene. A 197-bp band as the WT allele and a 115-bp band as the deleted allele were designed for genotyping by PCR amplification. The primers used are listed in **Supplementary file 7**.

### Generation of *Fbxo24*-HA-tagged transgenic mice

*Fbxo24*-HA-tagged transgenic mice were also generated by CRISPR/Cas9 technology and haploid embryonic stem cell microinjection. The sgRNAs of the C-terminus of the target gene *Fbxo24* (AAGA AGAGGGGTTCAGCTCT) were synthesized, annealed, and ligated to the pX330-mCherry plasmid. For the construction of the HA-tag DNA donor, the sequences encoding the left homologous arm, the HA tag, and the right homologous arm were amplified and ligated to the linear pMD19T. DKO-AG-haESCs were transfected with CRISPR-Cas9 plasmid and HA tag DNA donor. At 24 hr after transfection, the mCherry-positive haploid cells were enriched. In 7–8 days after plating, single colonies were

picked up, and positive colonies for the HA tag were selected by genomic DNA PCR amplification for Sanger sequencing. DKO-AG-haESCs with *Fbxo24*-C-HA were arrested in M-phase by culturing in a medium containing 0.05 μg/ml demecolcine for 10 hr and then used for intracytoplasmic injection as described previously (*Zhong et al., 2015*). Intracytoplasmic AG-haESC injection (ICAHCI) embryos were cultured in KSOM medium for 24 hr to reach the two-cell stage. 15–20 two-cell embryos were transferred into each oviduct of pseudo-pregnant females to produce *Fbxo24*-C-HA semi-cloned mice (F0). Then, the F0 female mice were crossed with WT male mice to generate heterozygous *Fbxo24*-C-HA F1 male mice for the experiments.

## Antibodies

The details of all commercial antibodies used in this study are presented in *Supplementary file 8*.

## Electron microscopy

For TEM, samples were fixed in 4% paraformaldehyde (PFA) containing 0.05% glutaraldehyde in 0.1 M phosphate-buffered saline (PBS), then post-fixed in 1% osmium tetroxide. Dehydration was carried out in ethanol (25, 50, 75, and 95%), and the samples were embedded in Epon 812. Ultra-thin sections with 80 nm thickness were prepared by ultramicrotome (Leica UC7, Leica Biosystems, Germany). Ultrathin sections were counterstained with uranyl acetate and lead citrate and examined with a TEM (Hitachi HT7700, Japan). For SEM, the samples were fixed in 2.5% glutaraldehyde solution in 0.1 M PBS and collected on poly-L-lysine-coated glass coverslips, followed by post-fixed in osmium tetroxide and dehydrated in graded ethanol series. Then, the samples were subjected to critical point drying and coated with gold/palladium for observation with an SEM (AZteclive Ultim Max 100, UK).

## Sperm and histological analysis

The epididymis isolated from 8-week-old mice and sperm allowed to swim into M16 medium (Sigma-Aldrich), which was maintained at 37°C and 5% $CO_2$. After 30 min, the medium was collected and diluted for sperm number and motility assessment. To determine sperm percent motility, a 10-μl sample was loaded onto a clean slide glass and covered with a coverslip. Motility was graded according to the WHO criteria under positive phase contrast microscopy at a total magnification ×400 (*Komori et al., 2006*). The length of midpiece was measured from the sperm neck to the sperm annulus by MitoTracker staining. All results were obtained from experiments performed on at least 6 mice per genotype group and 100 sperm per mouse. Testes and epididymis from WT and KO mice were fixed in Bouin's fixative and embedded in paraffin. For the histological analysis, sections of 5 μm were cut and stained with periodic acid-Schiff (PAS) or hematoxylin and eosin (H&E) after dewaxing and rehydration.

## Immunofluorescence

Testes were fixed in 4% PFA in PBS overnight at 4°C and then were sequentially soaked in 5, 10, 12.5, 15, and 20% sucrose in PBS and embedded in Tissue-Tek O.C.T. compound (Sakura 4583, Sakura Finetek USA, Inc, Torrance, CA) on dry ice. 5 μm cryosections were cut and washed with PBS three times (10 min per wash), then the cryo-sections were microwaved in 0.01 M sodium citrate buffer (pH 6.0) and cooled down to room temperature (RT) for antigen retrieval. After washing with PBS three times, the sections were blocked in a blocking solution (containing 3% normal donkey serum and 3% fetal bovine serum in 1% bovine serum albumin) for 1 hr. Then, the sections were incubated with primary antibodies overnight at 4°C and then incubated with secondary antibodies for 1 hr at RT. After washing with PBS and stained with DAPI, the sections were photographed under FluoView 1000 microscope (Olympus, Japan) with a digital camera (MSX2, Micro-shot Technology Limited, China). Images were merged using Adobe Photoshop (Adobe Systems, San Jose, CA).

## TUNEL staining

Testes were fixed in 4% PFA, embedded in Tissue-Tek O.C.T. compound, and cut section with 5 μm thick. Sperm were fixed in 4% PFA and permeated by 0.3% Triton X-100. TUNEL staining was performed using the TUNEL ApoGreen Detection Kit (YEASEN, 40307ES20, China). Images were obtained with a FluoView 1000 microscope (Olympus, Japan).

## Western blot

HEK293T cells (Cat# GNHu43, obtained from Stem cell Bank of Chinese Academic Science, the identity has been authenticated using Short Tandem Repeat profiling) were tested for mycoplasma contamination and showed negative result. Fresh mouse adult testes or HEK293T cells were collected, and proteins were extracted by using RIPA (Radioimmunoprecipitation assay) buffer (Beyotime, P0013J, China). In total, 40 µg of protein lysates were separated on a 10% sodium dodecyl sulfate–polyacrylamide gel electrophoresis (SDS–PAGE) gel, proteins were transferred to PVDF (Polyvinylidene Fluoride) membranes (Bio-Rad) and the membranes were blocked in 5% non-fat milk (blocking solution) for 1 hr. Primary antibodies were incubated overnight at 4°C after blocking. The membranes were washed with TBST three times and then incubated with a secondary antibody for 1 hr before using Luminol/enhancer solution and Clarity Western ECL Substrate (Bio-Rad Laboratories, Inc US). Western blot images were scanned by using Gel Doc XR system (Bio-Rad Laboratories, Inc US).

## IP assays

For testis tissue IP experiments, the testes were dissected and lysed in IP buffer (Beyotime, P0013J, China), clarified by centrifugation at 12,000 × g, and then pre-cleared with anti-HA nanobody agarose beads (AlpaLife, ktsm1305, China). The lysate was incubated with primary antibodies overnight at 4°C on a rotator and conjugated with anti-HA beads. The beads were washed with IP buffer and then boiled in 2× SDS loading buffer for western blotting analysis.

For cell line IP experiments, the HEK293T cells were transfected with indicated plasmids using Lipofectamine 2000 (Life Technologies). After 48 hr, IP was performed as described previously (*Li et al., 2022*). 25 µl mCherry-Trap bead 50% slurry (AlpaLife, ktsm1331, China) was used and all wash steps were performed with washing buffer (10 mM Tris–HCl pH 7.5, 150 mM NaCl, 0.5 mM Ethylenediaminetetraacetic Acid). mCherry-Trap beads were washed with dilution buffer prior to addition to the cell lysate. Beads were incubated with cell lysate at 4°C for 2 hr following another wash step. To elute the proteins from the beads, 40 µl sample buffer (120 mM Tris–HCl pH 6.8, 20% glycerol, 4% SDS, 0.04% Bromophenol blue, 10% β-mercaptoehanol) was added and boiled at 95°C for 5 min. mCherry-tagged proteins were detected by western blotting using an anti-mCherry antibody.

## Spermatogenic cell isolation

Spermatogenic cells were isolated from the whole mouse testis using the STA-PUT method described previously with slight modifications (*Bellvé, 1993*). In brief, testes were collected from 8-week-old mice WT and *Fbxo24* KO mice and decapsulated with collagenase treatment to remove Leydig cells. The dispersed seminiferous tubules were then digested with trypsin and DNase I to single-cell suspensions. Next, Sertoli cells were separated from germ cells by filtration through a 40-mm cell strainer and by adhesion to lectin-coated culture plates. Store the cells in the 1× Krebs buffer on ice. Different germ cell populations were separated using a manually prepared 0.5–5% discontinuous BSA (Bovine Serum Albumin) density gradient for velocity sedimentation sediment. After sedimentation (1.5 hr, 4°C), enriched fractions of the cell were manually collected, and the cell concentration was determined.

## RNA isolation and quantitative qPCR

Total RNAs were extracted from purified germ cell fractions using TRIzol reagent (Invitrogen, UK) following the manufacturer's procedure. The purity and concentration of RNA samples were determined using a Nanodrop ND-2000 spectrophotometer (Thermo Scientific, Madison, USA). Reverse transcription of 500 ng purified total RNA was performed by PrimeScript RT reagent kit with gDNA Eraser (Takara, Dalian, China). RT-qPCR was performed with SYBR green master mix (Takara, Dalian, China) on the ABI Step One System (Applied Biosystems) according to the manufacturers' procedure. The relative gene expression was quantified using the comparative cycle threshold method, with the *Gapdh* expression used for normalization.

## RNA-seq analysis

1 µg of total RNA was used from each group to prepare the mRNA libraries using TruSeq Stranded mRNA Library Preparation Kit Set A (Cat. No. RS-122-2101, Illumina) according to the manufacturer's instructions. All libraries were sequenced using the Illumina HiSeq 4000 platform. The FASTX-Toolkit

was used to remove adaptor sequences, and low-quality reads from the sequencing data. To identify all the transcripts, we used Tophat2 and Cufflinks to assemble the sequencing reads based on the UCSC mm10 mouse genome. The differentially expression analysis was performed by Cuffdiff. The differential expressed genes were set with the threshold of $p < 0.05$ and fold change >2.

### Small RNA-seq and annotation

For small RNA-seq, the total RNA from adult testes was gel fractionated, and those 16–40 nt in length were enriched by polyacrylamide gel electrophoresis, and purified for small RNA-seq. The small RNA libraries were constructed using the Digital Gene Expression for Small RNA Sample prep kit (Illumina). Sequencing of the small RNA library was performed by Illumina HiSeq Xten. Small RNA annotation was performed as described previously (*Kuramochi-Miyagawa et al., 2008*).

### Statistical analysis

All data are presented as mean ± standard deviation unless otherwise noted in the figure legends. Statistical differences between datasets were assessed by one-way analysis of variance or Student's *t*-test using the GraphPad Prism 8 software. p-values are denoted in figures by *$p < 0.05$; **$p < 0.01$; ***$p < 0.001$.

## Acknowledgements

This work was supported by National Natural Science Foundation of China (82001620, 82371627 to ZL and 82171605 to SY), the National Key R&D Program of China (2018YFC1004500 to LZ), the Open Research Fund of the National Center for Protein Sciences at Peking University in Beijing (KF-202205 to ZL), and the Open Research Fund of Key Laboratory of Reproductive Medicine of Guangdong Province (2020B1212060029 to ZL).

## Additional information

### Funding

| Funder | Grant reference number | Author |
|---|---|---|
| National Natural Science Foundation of China | 82001620 | Zhiming Li |
| National Natural Science Foundation of China | 82171605 | Shuiqiao Yuan |
| National Key Research and Development Program of China | 2018YFC1004500 | Liquan Zhou |
| Open Research Fund of the National Center for Protein Sciences at Peking University in Beijing | KF-202205 | Zhiming Li |
| Open Research Fund of Key Laboratory of Reproductive Medicine of Guangdong Province | 2020B1212060029 | Zhiming Li |
| National Natural Science Foundation of China | 82371627 | Zhiming Li |

The funders had no role in study design, data collection, and interpretation, or the decision to submit the work for publication.

### Author contributions

Zhiming Li, Conceptualization, Data curation, Software, Formal analysis, Supervision, Funding acquisition, Validation, Investigation, Methodology, Writing – original draft; Xingping Liu, Formal analysis, Investigation; Yan Zhang, Yuanyuan Li, Investigation; Liquan Zhou, Supervision, Funding acquisition,

Validation, Investigation; Shuiqiao Yuan, Conceptualization, Funding acquisition, Writing – review and editing

### Author ORCIDs
Liquan Zhou (iD) https://orcid.org/0000-0002-9332-9408
Shuiqiao Yuan (iD) https://orcid.org/0000-0003-1460-7682

### Ethics
All the animal procedures were approved by the Institutional Animal Care and Use Committee protocols (Permit Number: S2795) of Tongji Medical College, Huazhong University of Science and Technology, and the mice were housed in the specific pathogen-free facility of Huazhong University of Science and Technology. All experiments with mice were conducted ethically according to the Guide for the Care and Use of Laboratory Animal guidelines.

Reviewer #1 (Public Review): https://doi.org/10.7554/eLife.91666.3.sa1
Reviewer #2 (Public Review): https://doi.org/10.7554/eLife.91666.3.sa2
Reviewer #3 (Public Review): https://doi.org/10.7554/eLife.91666.3.sa3
Author Response https://doi.org/10.7554/eLife.91666.3.sa4

## Additional files

### Supplementary files
• Supplementary file 1. Identification of FBXO24 interactors by mass spectrometry.

• Supplementary file 2. Differentially expressed genes in the round spermatids between *Fbxo24* KO and wild-type (WT) mice by RNA-seq.

• Supplementary file 3. Gene Ontology (GO) enrichment of the down-regulated genes in the round spermatids of *Fbxo24* KO by RNA-seq.

• Supplementary file 4. Differentially expressed miRNAs in the testis between *Fbxo24* KO and wild-type (WT) mice by small RNA-seq.

• Supplementary file 5. Differentially expressed piRNAs in the testis between *Fbxo24* KO and wild-type (WT) mice by small RNA-seq.

• Supplementary file 6. The expression of transposable element-derived piRNAs in small RNA-seq data.

• Supplementary file 7. Primer sequences are used in this study.

• Supplementary file 8. Antibodies used in this study.

• MDAR checklist

### Data availability
RNA-seq and small RNA-seq data are deposited in the NCBI SRA database with the accession number PRJNA878933 and PRJNA878953, respectively.

The following datasets were generated:

| Author(s) | Year | Dataset title | Dataset URL | Database and Identifier |
|---|---|---|---|---|
| Li Z | 2022 | RNA-seq data of FBXO24 KO mouse | https://www.ncbi.nlm.nih.gov/bioproject/?term=PRJNA878933 | NCBI BioProject, PRJNA878933 |
| Li Z | 2022 | Small RNA-seq of FBXO24KO mouse | https://www.ncbi.nlm.nih.gov/bioproject/?term=PRJNA878953 | NCBI BioProject, PRJNA878953 |

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
