## [Editor Report · eLife assessment]

This **important** study provides insights into the role of FBXO24 in controlling spermiogenesis and male fertility in mice. The mouse models used and the data are **convincing**. This paper will interest biomedical researchers working on reproductive biology and fertility control.

---

## [Referee Report · Reviewer #1 (Public Review)]

In this study, Li et al., report that FBXO24 contributes to sperm development by modulating alternative mRNA splicing and MIWI degradation during spermiogenesis. The authors demonstrated that FBXO24 deficiency impairs sperm head formation, midpiece compartmentalization, and axonemal/peri-axonemal organization in mature sperm, which causes sperm motility defects and male infertility. In addition, FBXO24 interacts with various mRNA splicing factors, which causes altered splicing events in Fbxo24-null round spermatids. Interestingly, FBXO24 also modulates MIWI levels via its polyubiquitination in round spermatids. Thus, the authors address that FBXO24 modulates global mRNA levels by regulating piRNA-mediated MIWI function and splicing events in testicular haploid germ cells.

This study is performed with various experimental approaches to explore and elucidate underlying molecular mechanisms for the FBXO24-mediated sperm defects during germ cell development. Overall, the experiments were designed properly and performed well to support the authors' observation in each part. In addition, the findings in this study are useful for understanding the physiological and developmental significance of FBXO24 in the male germ line, which can provide insight into impaired sperm development and male infertility.

In the revised manuscript, the authors address most of the concerns raised in the previous review. The following are representative remaining points.

• Quantification of the defective, vacuolar mitochondria (80%) and missing annulus (30%) was not shown in the figures or described in the results as well as in a few other figures.

---

## [Referee Report · Reviewer #2 (Public Review)]

Spermatogenesis describes a complex sequence of differentiation events that lead to the development of genetically distinct male germ cells. The final part of spermatogenesis is called spermiogenesis, in which spermatids differentiate into mature sperm by developing an acrosome and a motile flagellum, which are required for reaching and successfully penetrating the oocyte. This process of spermatogenesis is based on a coordinated regulation of gene expressions in round spermatids. In the current study, FBXO24 was identified as a highly expressed protein in human and mouse testis. To define its biological role in vivo, the authors generated genetically engineered Fbxo24 knockout and Fbxo24-HA-labeled transgenic mouse models.

To elucidate the causes of the observed sterility in Fbxo24-KO males, the authors performed molecular and phenotypic analyses that revealed aberrant histone retention, incomplete axonemes, oversized chromatoid bodies (CB), and abnormal mitochondrial coiling along the sperm flagella. These results support the causal role of the FBXO24 gene in sperm motility.

Furthermore, the authors carefully characterized by SEM, TEM and western blot analyses that deletion of FBXO24 leads to incomplete histone-to-protamine exchange and defective chromatin interaction during spermiogenesis. In addition to increased MIWI expression, the authors show that FBXO24 interacts with SCF subunits and mediates the degradation of MIWI via K48-linked polyubiquitination.

This is a solid work demonstrating the role of FBXO24 in modulating alternative mRNA splicing, MIWI degradation and normal spermiogenesis.

---

## [Referee Report · Reviewer #3 (Public Review)]

This work is carried out by the research group led by Shuiqiao Yuan, who has a long interest and significant contribution in the field of male germ cell development. The authors study a protein for which limited information existed prior to this work, a component of the E3 ubiquitin ligase complex, FBXO24. The authors generated the first FBXO24 KO mouse model reported in the literature using CRISPR, which they complement with HA-tagged FBXO24 transgenic model in the KO background. The authors begin their study with a very careful examination of the expression pattern of the FBXO24 gene at the level of mRNA and the HA-tagged transgene, and they provide conclusive evidence that the protein is expressed exclusively in the mouse testis and specifically in post-meiotic spermatids of stages VI to IX, which include early stages of spermatid elongation and nuclear condensation. The authors report a fully sterile phenotype for male mice, while female mice are normal. Interestingly, the testis size and the populations of spermatogenic cells in the KO mutant mice show small (but significant) reduction compared to the WT testis. Importantly, the mature sperm from KO animals show a series of defects that were very thoroughly documented in this work by scanning and transmission electron microscopy; this data constitutes a very strong point in this paper. FBXO24 KO sperm have severe defects in the mitochondrial sheath with missing mitochondria near the annulus, and missing outer dense fibers. Collectively these defects cause abnormal bending of the flagellum and severely reduced sperm motility. Moreover, defects in nuclear condensation are observed with faint nuclear staining of elongating and elongated spermatids, and reduction of protein levels of protamine 2 combined with increased levels of histones and transition protein 1. All the above are in line with the observed male sterility phenotype.

The authors also performed RNASeq in the KO animal, and found profound changes in the abundance of thousands of mRNAs; changes in mRNA splicing patterns were noted as well. This data reveals deeply affected gene expression patterns in the FBXO24 KO testis, which further supports the essential role that this factor serves in spermiogenesis. Unfortunately, a molecular explanation of what causes these changes is missing; it is still possible that they are an indirect consequence of the absence of FBXO24 and not directly caused by it.

The finding that Miwi protein levels are increased in the FBXO24 KO testis is an important point in this work, and it is in agreement with the observed increased size of the chromatoid body, where most of Miwi protein is accumulated in round spermatids. This finding is well supported with experiments performed in 293T cells showing that Miwi ubiquitination is FBXO24 dependent in this ectopic system. Moreover, the authors detect reduced ubiquitination of endogenous Miwi protein immunoprecipitated from FBXO24 KO testis. Consistent with an increase in Miwi protein levels, Miwi-sized piRNAs show increased abundance in total RNA from FBXO24 KO testis. It has been documented that Piwi proteins stabilize their piRNA cargo, so the increase in piRNA levels in 29-32 nt sizes is most likely not a result of altered biogenesis, but increased half-life of the piRNAs as a result of Miwi upregulation. piRNAs have been involved in the regulation of mRNAs in the post-meiotic spermatid, but it is unclear how increased Miwi protein and its piRNA cargo at the levels observed in this study contribute to the complete infertility phenotype of the FBXO24 KO male mice.

Therefore, a well-reasoned narrative on if and how the absence of FBXO24 as an E3 ubiquitin ligase is responsible for the observed mRNA and protein differential expression is largely absent. If FBXO24-mediated ubiquitination is required for normal protein degradation during spermiogenesis, protein level increase should be the direct consequence of genuine FBXO24 targets in the KO testis. Apart from Miwi, the possible involvement of ubiquitination was not shown for any other proteins that the authors found interact with FBXO24 such as splicing factors SRSF2, SRSF3, SRSF9, or any of the other proteins whose levels were found to be changed (reduced, thus the change in the KO is less likely due to absence of ubiquitination) such as ODF2, AKAP3, TSSK4, PHF7, TSSK6 and RNF8. Interestingly, the authors do observe increased amounts of histones and transition proteins, but reduced amounts of protamines, which directly shows that histone to protamine transition is indeed affected in the FBXO24 KO testis, consistent with the observed less condensed nuclei of spermatozoa. Could histones and transition proteins be targets of the proposed ubiquitin ligase activity of FBXO24, and in its absence, histone replacement is abrogated? Providing experimental evidence to address this possibility would greatly expand our understanding on why FBXO24 is essential during spermiogenesis.

---

## [Author Response]

The following is the authors’ response to the original reviews.

**Reviewer #1 (Public Review):**
However, there are several concerns to be explained more in this study. In addition, some results should be revised and updated.

Thank you for your comments. The concerns were addressed by the description and experiment.

Some results were revised and updated accordingly.

**Reviewer #2 (Public Review):**
The minor weakness of the study is inconsistent use of terminology throughout the manuscript, occasional logic-jump in their flow, and missing detailed description in methodologies used either in the text or Materials and Methods section, which can be easily rectified.

Thank you for your review. We have revised the manuscript and corrected errors according to your comments.

**Reviewer #3 (Public Review):**
Importantly, besides the Miwi ubiquitination experiment which is performed in a heterologous and therefore may not be ideal for extracting conclusions, the possible involvement of ubiquitination was not shown for any other proteins that the authors found that interact with FBXO24. Could histones and transition proteins be targets of the proposed ubiquitin ligase activity of FBXO24, and in its absence, histone replacement is abrogated?

Thank you for your comments. The histones and transition proteins were not found in the immunoprecipitates of FBXO24, suggesting they are not the direct targets of FBXO24, shown in Figure S3G.

Miwi should be immunoprecipitated and Miwi ubiquitination should be detected (with WB or mass spec) in WT testis.

We agree with this suggestion. In the revision, the expression and ubiquitination of MIWI were detected in WT testis by the immunoprecipitation and ubiquitination assay, as shown in Figure 8H.

Therefore, the claim that FBXO24 is essential for piRNA biogenesis/production (lines 308, 314) is not appropriately supported.

We appreciate the comment. We have revised the description and modified the claim on page 11.

Reviewing Editor's note for revision(1) As noted by all three reviewers, as currently written the rationale to focus on MIWI is not entirely clear. A transitional narrative to focus on MIWI needs to be provided as well as an explanation for how the absence of FBXO24 as an E3 ubiquitin ligase is responsible for the observed mRNA and protein differential expression.

We appreciate your comments. We have supplemented the transitional narrative by focusing on MIWI and explained mRNA and protein differential expression upon FBXO24 deletion, shown on Page 7 and Page 13, respectively.

(2) As it can be indirect, mass spec detection of MIWI in testis co-IP and MIWI ubiquitination should be detected (with WB or mass spec) in WT testis.

In the revision, the expression and ubiquitination of MIWI were detected in WT testis by the immunoprecipitation and ubiquitination assay, as shown in Figure 8H.

(3) Please tone down the claim that FBXO24 is essential for piRNA biogenesis/production as it requires further evidence.

We have revised the description and modified the claim on page 11.

(4) Ontology analysis of the genes with abnormally spliced mRNAs to provide an explanation for developmental defects.

In the revision, we have performed the ontology analysis and provided new data regarding the abnormally spliced genes, as shown in Figure S4D.

**Reviewer #1 (Recommendations For The Authors):**
Major comments(1) The authors performed mainly with the WT (or knock-in) and Fbxo24-knockout mouse model. Do the heterozygous males and their sperm have any physiological defects like FBXO24-deficient mice?

This is a good question. We did the phenotype analysis and found that heterozygous males are all fertile, and their sperm do not have any physiological defects.

(2) Fbxo24-KO sperm carries swollen mitochondria. How do the mitochondria affect sperm function?

Thank you for raising this interesting question. Based on our data and published literature, the defective mitochondria were associated with energetic disturbances and reduced sperm motility, as shown on Page 12.

(3) TEM images show that Fbxo24-KO spermatids carry swollen mitochondria and enlarged chromatoid bodies. How the swollen mitochondria and enlarged chromatid are defective for sperm motility and flagellar development, requires more explanation. In addition, it is unclear how the enlarged diameter of the chromatoid body is critical for normal sperm development.

Thank you for your comments. The chromatoid bodies are considered to be engaged in mitochondrial sheath morphogenesis. Analysis of the chromatoid bodies' RNA content reveals enrichment of PIWI-interacting RNAs (piRNAs), further emphasizing the role of the chromatoid bodies in post-transcriptional regulation of spermatogenetic genes. We added this explanation on Page 12-13.

(4) The authors only show band images to compare the protein amounts between WT and KO sperm and round spermatids. As the blots for loading controls are not clear, the authors should quantify the protein levels and perform a statistical comparison.

We quantified the protein levels and performed a statistical comparison, as shown in Figure S3B.

(5) The authors show the defective sperm head structure from Fbxo24-KO sperm in Figure 5. However, the Fbxo24-KO sperm heads seem quite normal in Figure 3. How many sperm show defective sperm head structure? In addition, the authors observed altered histone-to-protamine conversion in sperm, but it is unclear whether the altered nuclear protein conversion causes morphological defects in the sperm head.

We appreciate the comments. In our study, we found over 80% of Fbxo24 KO sperm showed defective structure in the sperm head. Altered histone-to-protamine conversion caused the decondensed nucleus of Fbxo24 KO sperm. Notably, in many knockout mice studies, impaired chromatin condensation is frequently associated with abnormal sperm head morphology, as shown in reference 15 of Page 8.

(6) The authors compare the protein levels of RNF8, PHF7, TSSK6, which participate in nuclear protein replacement in sperm. However, considering the sperm is the endpoint for the nuclear protein conversion, it is unclear to compare the protein levels in mature sperm. The authors might want to compare the protein levels in developing germ cells.

Thank you for your comment. Yes, we actually detected the protein levels of RNF8, PHF7, and TSSK6 in the testes, not in sperm. We have corrected it in the Figure 5E. We apologize for our carelessness.

(7)This reviewer suggests describing more rationales for how the authors focus on the MIWI protein. Also, it is wondered whether MIWI is also detected from testis co-IP mass spectrometry.

We agree with this suggestion. Since MIWI was a core component of CB and also identified as an FBOX24 interacting partner from our immunoprecipitation-mass spectrometry (IP-MS) (Table S1), we focused on the examination of MIWI expression between WT and Fbxo24 KO testes. We have added this description in the revision (see lines 191-193 on page 7).

(8) The authors need to provide a more detailed explanation for how the altered piRNA production affects physiological defects in germ cell development. In addition, it will be good to describe more how the piRNAs affect a broad range of mRNA levels.

Thank you for your comments. The previously published studies have demonstrated that piRNAs could act as siRNAs to degrade specific mRNAs during male germ cell development and maturation. We have cited these studies on lines 369-372 of Page 13.

(9) The authors observed an altered splicing process in the absence of FBXO24. However, it is a little bit confusing how the altered splicing events affect developmental defects. Therefore, the authors should state which mRNAs have undergone abnormal splicing processes and provide ontology analysis for the genes.

We have performed the ontology analysis and showed the new data in Figure S4D.

Minor comments(1) Figure 1A-C - Statistical comparison is missed. Numbers for biological replication should be described in corresponding legends.

Thank you for your careful review. We have provided the statistical comparison and the numbers for biological replication in the legends of Figure 1A-C.

(2) Figure 1E, F - Current images can't clearly resolve the nuclear localization of the FBXO24 testicular germ cells. To clarify the intracellular localization, the authors should provide images with higher resolution.

The resolution of Figure 1E, F was improved, as suggested. Thank you!

(3) Figure 1E, F - Scale bar information is missing.

The scale bars of Figure 1E, F were provided.

(4) It will be much better to show the predicted frameshift and early termination of the protein translation in Fbxo24-knockout mice.

The predicted frameshift of Fbxo24-knockout mice was added and shown in Figure S1B.

(5) It is required to provide primer information for qPCR.

The primer information for qPCR was provided, as shown in Table S7.

(6) The authors describe that Fbxo24-KO sperm show abrupt bending of the tail. However, the description is unclear and the sperm shown in Figure 3C seems quite normal. The authors should clarify the abnormal bending pattern of the tail and show quantified results.

Thank you for pointing out this issue. In Fbxo24 KO sperm, abnormal bending of the sperm tails mainly included neck bending and midpiece bending. We have shown them in Figure S3A.

(7) The authors mention that Fbxo24-KO sperm have swollen mitochondria at the midpiece, but this is also unclear. How many mitochondria are swollen in Fbxo24-KO sperm?

This is a good question. However, since it is very difficult to observe all of the mitochondria in each sperm using the electronic microscope, we could not quantify the swollen mitochondria in Fbxo24 KO sperm.

(8) Scale bar information is missed - Fig 3C insets, Fig 3D, Fig 3F insets, 4A insets, Figure 4C insets.

All the scale bars have been added.

(9) How many sperm have annulus defects? In Figure 3F, WT sperm does not have an annulus, which could be damaged during sample preparation. Is the annulus defects in Fbxo24-KO sperm consistent?

Thank you for asking these questions. Based on our results, about 30% of Fbxo24 KO sperm showed defective annulus structure. Since both TEM (Figure 3F) and SEM (Figure 3G) results clearly showed the defective annulus structure of Fbxo24 KO sperm, we believe the annulus defects are consistent and highly unlikely caused by sample preparation.

(10) A Cross-section image for the endpiece of Fbxo24-KO sperm is not suitable. There is a longitudinal column structure of the principal piece.

Thank you for your comments. It is difficult to observe a completely longitudinal structure of sperm tail under TEM. The cross-section of the endpiece and principal piece allowed us know the structure of the axoneme, ODFs and fibrous sheath (FS).

(11) The endpiece of Fbxo24-KO sperm seems to have a normal axoneme. Do all endpieces of Fbxo24KO sperm have normal axoneme? Also, the authors need to describe whether an axonemal structure is damaged and disrupted in all Fbxo24-KO sperm.

Our TEM data showed the axonemal structure was impaired in the endpiece of Fbxo24 KO sperm (See right panels of Figure 3H). Moreover, based on the ultrastructure analysis of TEM, we found over 90% of Fbxo24 sperm had a damaged axonemal structure.

(12) Reference blots in Fig 3I, 3J, 4E (left), 5C and 5E are quite faint. The authors should replace the blot images.

Thank you for pointing out this. We have rerun Western blot multiple times but could not obtain better images due to antibody sensitivity. However, we quantified the protein levels and performed a statistical comparison, as shown in Figure S3B, to establish a good readout from these images for the readers.

(13) Loading controls are required - 7D-H.

Done as suggested. Thanks!

(14) How do the authors measure the midpiece length? From where to where? This should be clarified.

Good question. We measured the midpiece length from the sperm neck to the sperm annulus by MitoTracker staining. We have clarified this on Page 16.

(15) How are the bands for Fbxo24 shifted during IP in Fig 7A?

The protein modification in the interaction may cause the band shift.

(16) There are several typos throughout the manuscript. Please check carefully and fix them.

Thank you for your careful review. We have corrected and fixed all the typos as far as we can.

**Reviewer #2 (Recommendations For The Authors):**
Major comments(1) Please provide a schematic of HA-Fbxo24 knock-in construct and strategy together with knockout (Figure S2) or even separately early in Figure S1. The description of using the transgenic mouse is mentioned even earlier than the knockout but there are no citations or methods provided in the text other than that listed in Materials and Methods.

Thank you for your suggestion. As suggested, the schematic of the HA-Fbxo24 knock-in strategy has been supplemented in Figure S2A. The description of using the transgenic mouse has been added to the results, as shown on page 4 of lines 102-103.

Also, it is not clear to what extent the phenotypic and molecular characterization of HA-transgenic mice is performed. For example, Lines 134-139: The use of Fbxo24-HA labeled transgenic mice results in the rescue of spermatogenesis and fertility as shown in Figure 2F by measuring the litter size. It is not clear how this observation leads the author to state that this rescues defects in spermiogenesis. Please clarify how and what other measures are taken to support this conclusion. Is the observed infertility due to defects in spermatogenesis or spermiogenesis?

Thank you for your question. We crossed FBXO24-HATag males with FBXO24−/− females to obtain FBXO24−/−; FBXO24-HATag males. We examined the testes volume and histological morphology of FBXO24−/−; FBXO24-HATag males and found that they were similar to FBXO24+/−; FBXO24-HATag littermates, indicating that spermatogenesis was restored, as shown in Figure S2H.

(2) Line 107 vs Line 114: Please use the terminology spermatogenesis and spermiogenesis consistently throughout the text. Earlier in the introduction, the authors clearly defined that spermatogenesis involves three phases, with the third phase referred to as spermiogenesis. However, the author concludes in the first line that "FBXO24 plays a role during spermatogenesis" while summarizing at the end of the paragraph that this protein is "expressed in haploid spermatids specifically during spermiogenesis". Therefore, it is not clear whether the authors conclude that FBXO24 is important for all of spermatogenesis (line 107) or only for part of spermiogenesis (line 114). Another example is line 219 vs. 238: At this point in the manuscript, it is again unclear whether the authors want to study molecular changes during spermatogenesis or spermiogenesis upon FBXO24 depletion. Many examples of such cases throughout the text, and it is recommended to be consistent in using more restrictive terminology whenever applicable for a clear interpretation.

We thank you for your careful review. We have double-checked the terminology of spermatogenesis and spermiogenesis and made it consistent throughout the text of the revised manuscript.

(3) It is not clear how rampant/frequent the Fbxo24-knockout sperm show defects in head morphology based on Figures 3C, 3F, and 5A since it seems that there are some sperm showing relatively normallooking sperm heads. Please provide quantification.

We have performed the quantification and found that over 80% of Fbxo24 KO sperm showed defective structures in the sperm head.

(4) Figure 3B: The authors describe in the figure legend that 3 mice were analyzed in each group. The standard deviation for the WT analysis is missing, or if the author wanted to set the WT value to 100%, the bar and scale shown on the y-axis do not fit. The value for WT looks more like 95%.

We have indeed analyzed sperm motility based on the WT value set at 100% and have revised Figure 3B in the revision. We apologize for this oversight.

(5) Figure 3 B and C: It is not clear how the motility is measured. Is CASA used (not described in Methods). The conclusion about abnormal flagellar bending in KO spermatozoa cannot be drawn from the static microscopic images alone. Please provide more details of motility analysis together with videos of live cell imaging.

The sperm motility was measured manually using a hemocytometer, according to the reference.

We provided the details of sperm motility analysis in the Materials and Methods section on Page 16.

(6) Figure 3 I and J: These are one of a few figures that are not supported by statistical analysis. In particular, for 3I, GAPDH controls of WT and KO protein do not show equal loading, which could explain the lower expression of the KO protein. Please show normalized bar graphs with multiple biological replicates or at least show a representee technical replicat that shows equal loading of GAPDH to better support the conclusion.

Thank you for your suggestion. Statistical comparison of relative protein expression was supplemented, as shown in new Figure S3B.

(7) Line 184: It is not clear how the authors define a swollen mitochondrion? Are there any size criteria (roundness) that can be measured to distinguish between a swollen and a non-swollen mitochondrion? It is recommended to use another terminology as often 'swollen' implies there is a difference in osmolarity but there is no experiment to support this implication.

Thank you for your comment. We have changed the “swollen” to “vacuolar” in the revision, as shown on Page 7.

(8) Figure S4, without a bright field image, it is hard to see the purity and morphology of the isolated prep. Please provide the bright field images together or as overlaid images.

We agree with your comment. We have provided the overlaid images in new Figure S4A.

(9) There is a big logic jump in what prompts the authors to look MIWI protein level and link the observation to MIWI/piRNA pathway in both Introduction and Results while it is one of the main findings. It is recommended to provide a better rationale and logical flow in the text.

Thank you for your suggestion. We have added a sentence explaining why we wanted to focus on studying MIWI expression (see lines 190-193 on page 7).

Minor comments(1) Please keep all the conventions of gene vs. protein nomenclature. For example, write the genes mentioned in the figures in italics with the first letter in Capital, as it is done in the main part. Proteins should be in ALL CAPITAL like FBXO24.

The names of gene and protein have been revised in the revision, as suggested.

(2) In the MM section, the name of the manufacturer and the location of the materials used are missing in several sections. Please go back through the MM section and add this information in the appropriate places.

Done as suggested. Thank you!

(3) On page 4, the authors mentioned that "Further qPCR analysis of developmental testes and purified testicular cells showed that FBXO24 mRNA was highly expressed in the round spermatids and elongating spermatids (Fig 1B-C)". Please include statistical analyses for Fig 1B-C as well as for Fig 1A to support the written statements.

Statistical comparison was supplemented, as shown in Figure 1. P-values are denoted in figures by *p < 0.05.

(4) Figure 3E: Please describe in more detail how the length of the midpiece was measured. Was it based on TEM images or based on fluorescent images using MitoTracker?

As we responded to Reviewer #1, we measured the midpiece length from the sperm neck to the sperm annulus by MitoTracker staining. We have clarified this in the Method and Material section on Page 16.

(5) Line 431: In the "Electron Microscopy" section of the MM part, the author should indicate the ascending ethanol series (%) used.

Done as suggested. Thank you!

(6) Line 432: The thickness of the sections prepared is missing, as well as an indication of the microtome used.

We have added thickness and the microtome in the Method and Material section on Page 16.

(7) Line 433: If the generated tiff files have been processed with Adobe Photoshop, this information is missing.

We have provided information on the usage of Adobe Photoshop for the generation of tiff files on Page 17.

(8) Lines 445, 452, 467: In some places in the paper, the temperature is written with a space between the number and {degree sign}C, and sometimes it is not. Please go through the paper and make it consistent. The usual spelling is 4{degree sign}C.

We have gone through the manuscript and checked all the spelling of temperature writing to make them consistent. Thank you for careful review.

(9) Line 469: The gel documentation system used is not mentioned.

Done as suggested. Thank you!

(10) Line 469: The 'TM' should be superscripted.

Done as suggested.

(11) Line 489: A space is missing between the changes and the parenthesis.

Done as suggested.

(12) Line 495-496: The authors write that the fractions enriched with round spermatids after sedimentation were collected manually. Was a determination of cell concentration - e.g., 2 x106 cells/ml -performed after collection of the cells? How were the cells stored until use? Please add the sedimentation time and used temperature.

Store the cell in the 1´ Krebs buffer on ice. The cell sediment was through a BSA density gradient for 1.5 h at 4°C. The cell concentration was determined after collection, as shown on Page 18.

(13) Line 505: spelling error. Instead of " manufacturer's procedure" it is written manufactures' instructions.

The spelling error was corrected.

(14) Line 520: Please write a short sentence on how the purification of the 16-40 nt long RNA was performed.

The length of 16–40 nt RNA was enriched by polyacrylamide gel electrophoresis. We added this information on Page 19 of line 531.

(15) Line 528: The version of the used GraphPad software is missing.

The version of GraphPad software was supplemented, as shown on Page 19.

(16) Line 677: For qPCR analyses, the number of mice analyzed (N) and a statistical evaluation are missing.

The statistical comparison and the numbers for biological replication were added, as shown on Page 26.

(17) Figure 3D: Please add a scale bar.

Done as suggested. Thanks!

(18) Line 371 and Line 377: Two times "in summary" is written. Please make one summary for the whole paper.

This sentence was revised, as shown in Page 13.

(19) Line 382: To be consistent in the whole paper, please write Figure 10 in bold letters.

Done as suggested.

(20) Please make the size and font of the references consistent with the main text.

Done as suggested. Thanks again for your careful review.

**Reviewer #3 (Recommendations For The Authors):**
I would like to see the description of the FBXO24 immunoprecipitation experiment performed in HEK293T cells. This somatic cell line does not normally express Miwi, so how Miwi was detected in FBXO24 mCherry IP beads? It is not mentioned if Miwi is expressed from a recombinant vector in this experiment. Similarly, I would like to see a better description of the experiment described in the same paragraph towards the end of it with the ubiquitin peptides, it is not clear.

Thank you for your comments. FBXO24-mCherry was expressed in HEK293T cells and the immunoprecipitates was incubated with the protein lysate of the testes (see lines 268-272 on Page 10). The description of the ubiquitin experiment was added as well, as shown in lines 283-286 on Page 10.

Line 263: I think the term ectopic here is not appropriate, a correction is needed.

We have changed “ectopic” to “increased” in the revision (see line 268 on Page 10).

I would like the authors to provide a tentative explanation or evidence of why FBXO24 KO males are completely sterile, even though there are still mature sperm produced with some motility. Since there are defects in nuclear condensation it will be very relevant to check DNA damage/fragmentation, which could contribute to the sterility phenotype.

This is a good suggestion. We reanalyzed the sperm DNA damage by TUNEL staining and shown the new data in Figure S3E-F.

Line 213: There have been some conflicting reports about the role of RNF8 in spermiogenesis, but a recent report has shown that RNF8 is not involved in histone PTMs that mediate histone to protamine transition (Abe et al Biol Reprod 2021).

Thank you for your comment. We have cited this critical reference and discussed it in Discussion section on Page 12.

Figure 7: I would like to see zoomed-out views of the affected exons, so that flanking unaffected exons can be used as a reference for unaffected splicing. Most of the genome browser views in this image only show affected exons and it is impossible to see if these alone are affected or if the reduced RNAseq coverage in those exons is a result of overall reduced mapped reads in these genes. Also, a fixed Y axis with the same max value should be shown for these genome browser snapshots so that the expression level is comparable between the two genotypes.

Thank you for your comments. Loading control of RT-PCR and scale range of Y axis were added in new Figure 7.

Minor corrections:Line 70: correct "..functions as protein-protein interaction..".

Thank you for your careful review. We have corrected this sentence (see line 69 on Page 3).

Line 101: correct "..qPCR analysis of developmental testis..".

We have corrected this sentence (see line 100 on Page 4). Thanks again.

Line 116: correct "..results in detective..".

Corrected.

Line 186: correct ".. explored..".

Corrected.

Line 218: correct ".. gene expressions.

Corrected.

Line 221: correct "..genes significantly differentiated expressed".

Corrected.

Line 241: FBXO24 was shown earlier in both cytoplasm and nucleus.

We have changed “FBXO24 is mainly confined to the nucleus” to “FBXO24 expressed in the nucleus”, as shown in line 247 on Page 9.

Line 501-502: correct "..reverse transcriptional".

“reverse transcriptional” was changed into “reverse transcription”, showing in Page 18.

Line 686: correct ".. deficiency male..".

Corrected.

Line 769: correct "..Western blots were adopted..".

Corrected.

Line 784: correct "..WT tesis..".

Corrected.

I cannot understand exactly what is shown in Figure 9B. Some elements marked on the X-axis are single base locations (-2K, TSS, +2K) and others are stretches of sequences so they cannot be equivalent. Why there is only an intron shown? There should be a measure of normalized expression on the Y-axis.

Thank you for your questions. The X-axis means that genome segments were scaled to the same size and were calculated the signal abundance, which was analyzed by computeMatrix. Aim to know the piRNA source, piRNA was mapped to the gene body, including introns, CDS and UTRs. The value of the Y-axis is the normalized count.

Figure 6F is not needed.

Figure 6F was used to illustrate the number of different types of mRNA splicing upon FBXO24 deletion in the round spermatids. To better understand the splicing for the reader, we decided to keep it.

The last two paragraphs of the discussion seem to be redundant.

Thank you for pointing out this. We have revised the last two paragraphs of the discussion.